# Continuously Updating Digital Twins using Large Language Models

**Harry Amad** [1]   **Nicolás Astorga** [1]   **Mihaela van der Schaar** [1]

## Abstract

Digital twins are models of real-world systems that can simulate their dynamics in response to potential actions. In complex settings, the state and action variables, and available data and knowledge relevant to a system can constantly change, requiring digital twins to continuously update with these changes to remain relevant. Current approaches struggle in this regard, as they require fixed, well-defined modelling environments, and they cannot adapt to novel variables without re-designs, or incorporate new information without re-training. To address this, we frame digital twinning as an in-context learning problem using large language models, enabling seamless updates to the twin at inference time. We develop CALM-DT, a Context-Adaptive Language Model-based Digital Twin that can accurately simulate across diverse state-action spaces using in-context learning alone by utilising fine-tuned encoders for sample retrieval. We empirically demonstrate CALM-DT's competitive performance with existing digital twin approaches, and its unique ability to adapt to changes in its modelling environment without parameter updates.

## 1. Introduction

**What is a digital twin?** Digital twins (DTs) are computational models that simulate real-world system dynamics. They have been applied in a variety of fields, such as finance (Slepneva et al., 2021), climate science (Voosen, 2020), manufacturing (Rosen et al., 2015), and medicine (Katsoulakis et al., 2024), and they are particularly useful for scenario planning, by modelling how a system will respond to various actions (Tao et al., 2018; Corral-Acero et al., 2020). For example, consider a medical patient with oropharyngeal carcinoma, for whom deciding whether to sequentially or concurrently administer chemo- and radiotherapy is non-trivial (Cooper et al., 2004; Pignon et al., 2009). A DT can model factors related to their disease, such as tumour volume, and how they respond to certain interventions to assess the efficacy of different treatment plans, leading to optimal results (Tardini et al., 2022).

**Why must a digital twin be able to continuously update?** An important aspect of an effective DT is its ability to continuously learn and update as time passes (Tzachor et al., 2022; National Academy of Engineering and National Academies of Sciences, Engineering, and Medicine, 2024). DTs operate in user-defined *modelling environments*, that dictate the state and action variables they can model, and the data and knowledge bases from which they can derive insights. In *dynamic* settings, however, the variables that best describe a system, the actions that apply to it, and the pertinent surrounding information can constantly evolve, and models become obsolete if they cannot easily adjust to new conditions (Lu et al., 2018). Consider an illustrative example of using DTs to guide care for patients with cystic fibrosis (CF). Medical practice can evolve rapidly,[1] exemplified by the 2012 approval of Ivacaftor, a transformative therapy for CF patients with specific gene mutations (Ramsey et al., 2011). Such breakthroughs demand immediate integration into clinical workflows, and if a CF patient's DT cannot easily adapt to incorporate novel treatments like Ivacaftor, it can quickly lose relevance. Furthermore, given the rarity of CF, there is limited patient data available, and therefore any new information, such as annual data releases from the UK CF registry,[2] can be highly impactful. DTs that cannot continuously learn from new information will therefore perform increasingly sub-optimally for CF patients over time.

**Why do current approaches struggle to continuously update?** Existing DTs typically involve *knowledge-driven mechanistic components* (Laubenbacher et al., 2024) defined by expert-derived equations, *data-driven machine learning (ML) components* (Kreuzer et al., 2024) defined by neural networks, or some *hybrid* combination of these (Sokolov et al., 2021; Holt et al., 2024). For these approaches, updating modelling environments can be laborious. Updating

[1]Department of Applied Mathematics and Theoretical Physics, University of Cambridge, Cambridge, United Kingdom. Correspondence to: Harry Amad <hmka3@cam.ac.uk>.

*Proceedings of the 42nd International Conference on Machine Learning*, Vancouver, Canada. PMLR 267, 2025. Copyright 2025 by the author(s).

---

[1]The FDA approved 50 novel drugs in 2024 (Food and Drug Administration, 2024).

[2]https://www.cysticfibrosis.org.uk/

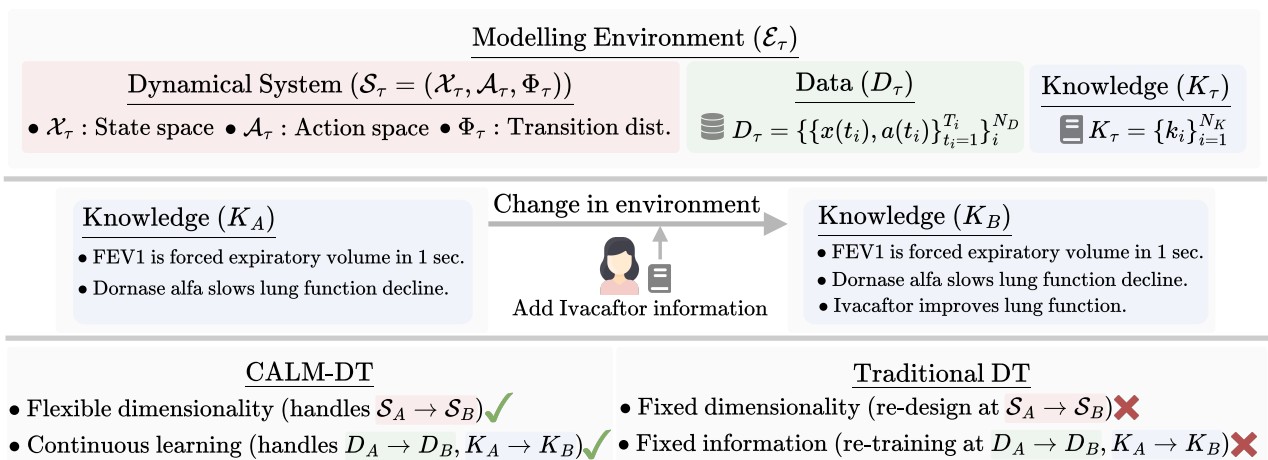

Figure 1. DTs operate within a user-defined modelling environment $\mathcal{E}_\tau$, with a certain abstraction of the target system $\mathcal{S}_\tau$ with state space $\mathcal{X}_\tau$ and action space $\mathcal{A}_\tau$, as well as relevant data $D_\tau$ and knowledge $K_\tau$. Modelling environments change when the user alters $\mathcal{S}_\tau$ to incorporate additional state or action variables, or when new data or knowledge is added to $D_\tau$ or $K_\tau$ (example addition to $K_\tau$ shown in middle panel). Traditional DTs cannot adapt to changes in $\mathcal{E}_\tau$ without re-design and re-training, while our CALM-DT can.

mechanistic components to accommodate new variables or information requires significant expert involvement to re-define the equations underpinning the model, especially in complex settings. Similarly, ML components operate with fixed state-action dimensions and training data, and they cannot incorporate novel variables or information without undergoing potentially expensive re-training.

For a DT to remain relevant in real-world settings it must allow inputs and outputs of *flexible dimensionality*, such that it can incorporate new variables without re-design, and be able to *continuously learn* and leverage new information without re-training.

**How can large language models address this?** Large language models (LLMs) acquire domain knowledge during pre-training, which can be augmented with task-specific knowledge and data at inference time through prompting. As such, LLMs can be novelly framed as an instantiation of a hybrid DT, incorporating both knowledge- and data-driven insights into simulation. Importantly, LLMs operate on natural language, permitting inputs of flexible and arbitrary dimensionality, and their in-context learning abilities allow new information to be incorporated without re-training (Brown et al., 2020).

While not their original use case, LLMs have shown promise in dealing with time-series data (Xue & Salim, 2023; Jin et al., 2024), most notably in an entirely zero-shot fashion (Gruver et al., 2024), suggesting that their use as DTs is plausible. However, DTs must be able to simulate across potentially vast state-action spaces, and providing enough data to allow an LLM to achieve this with in-context learning alone can be challenging. LLM context windows are finite, and performance can degrade with excessive context length (Liu

et al., 2024b). This encourages us to develop a simulation strategy that manages context length by making intelligent adjustments to the supplied context mid-generation.

In this work, we investigate the ability of LLMs to act as hybrid DTs, making the following contributions:

1. We identify incompatibilities of existing DTs with dynamic modelling environments, and establish a set of desiderata for DTs in such conditions (§2). We show that an LLM-based DT can satisfy these desiderata (§3).

2. We propose **CALM-DT**: a **C**ontext-**A**daptive **L**anguage **M**odel-based **D**igital **T**win that can adapt to changes in its modelling environment without re-design or re-training (§4). We address problems with excessive context length by adjusting the information supplied to the LLM mid-generation, to ensure maximum relevance to the current simulation state. We select trajectories from related systems to include in context using a fine-tuned bi-encoder framework, retrieving samples that are expected to minimise the LLM generation error at the current simulation state. During retrieval, we append LLM-generated summaries of trajectory trends to encoder inputs to improve the identification of temporal patterns.

3. We empirically demonstrate that CALM-DT outperforms existing DTs, and we showcase its unique ability to remain relevant across changes in modelling environments without parameter updates (§6).

Throughout this work, we will continually refer back to our CF DT example for illustrative explanations.

## 2. Digital Twins in Dynamic Environments

A dynamical system is defined as a three-tuple $\mathcal{S} := (\mathcal{X}, \mathcal{A}, \Phi)$, where $\mathcal{X} \subseteq \mathbb{R}^{d_\mathcal{X}}$ is a $d_\mathcal{X}$-dimensional state space, $\mathcal{A} \subseteq \mathbb{R}^{d_\mathcal{A}}$ is a $d_\mathcal{A}$-dimensional action space, and $\Phi$ is a transition distribution. The system at time $t \in \mathcal{T}$ has history $h(t) = \{(x(k), a(k))\}_{k=0}^{t-1} \in \mathcal{H} = (\mathcal{X} \times \mathcal{A})^*$ of $t$ state-action pairs, where $x(k) \in \mathcal{X}$ is a state vector and $a(k) \in \mathcal{A}$ is an action vector. The dynamics of the system is described by the transition $\Phi : \mathcal{H} \times \mathcal{T} \to P(\mathcal{X})$, and actions are determined by an external policy distribution $\pi : \mathcal{H} \times \mathcal{X} \times \mathcal{T} \to P(\mathcal{A})$.

When constructing a DT, a user defines a modelling environment $\mathcal{E}_\tau := (\mathcal{S}_\tau, D_\tau, K_\tau)$ which contains an abstraction $\mathcal{S}_\tau = (\mathcal{X}_\tau, \mathcal{A}_\tau, \Phi_\tau)$ of the system for the DT to approximate, a dataset $D_\tau = \{\{x_i(t), a_i(t)\}_{t=0}^{T_i-1}\}_{i=1}^{N_D}$ of observed histories from related systems, and a knowledge base $K_\tau = \{k_i\}_{i=1}^{N_K}$ of descriptions (e.g. physical laws) to guide modelling. Crucially, modelling environments are dynamic, as users may wish to redefine $\mathcal{S}_\tau$ and expand $D_\tau$ or $K_\tau$ at any moment, inducing a *change in modelling environment* $\mathcal{E}_\tau \to \mathcal{E}_{\tau+1}$. For example, the approval of the novel drug Ivacaftor in 2012 could lead to a user inducing a change in modelling environment by expanding $\mathcal{A}_\tau$ to incorporate it, and adding to $D_\tau$ and $K_\tau$ to describe its effects.

For a DT to remain relevant to a dynamical system in real-world settings, it must be able to easily incorporate additional state and action variables and refine its approximation with newly available information. To do so, a DT must satisfy the following desiderata:

[D1] **Flexible dimensionality.** To incorporate new state and action variables as $\mathcal{S}_\tau$ evolves, a DT must allow state-action inputs and outputs of varying and arbitrary dimensionality.

[D2] **Continuous learning capability.** To leverage updates to $D_\tau$ and $K_\tau$, a DT must be able to integrate new information without parameter updates.

Existing DT approaches do not satisfy these desiderata, since they have inflexible architectures that are fixed at design time. Upon $\mathcal{E}_\tau \to \mathcal{E}_{\tau+1}$ they require architectural overhauls to accommodate new $\mathcal{X}_{\tau+1}, \mathcal{A}_{\tau+1}$ dimensions and full re-training on $D_{\tau+1}$ and $K_{\tau+1}$. This necessitates a shift towards DTs that natively support dynamic modelling environments, a challenge we address through LLM-based simulation.

## 3. LLMs as Digital Twins

To develop DTs that remain relevant in dynamic modelling environments, we leverage LLMs, since they naturally fulfil [D1-2] from §2. Unlike traditional methods that fix $\mathcal{S}_\tau$, $D_\tau$

and $K_\tau$ at design time, LLMs can adapt to evolving state-action spaces, and learn from new information without re-designs or parameter updates. LLMs operate on free-form natural language, decoupling them from rigid state-action schemas. An LLM-based DT will therefore satisfy [D1] if a mapping $g$ is developed from state-action pairs of arbitrary dimensionality to natural language. Furthermore, LLMs have the capacity to learn at inference time with in-context learning (Brown et al., 2020). They therefore satisfy [D2], as their simulations can incorporate new information without parameter updates through changes to their prompts.

There are outstanding practical challenges with LLM-based DTs, however. The dimensionality of the state-action space for complex systems can be large, and their dynamics can vary significantly in this space. DTs can therefore require large amounts of data to accurately approximate system dynamics across these large spaces. The amount of information that LLMs can effectively incorporate at inference time is limited, however, making it infeasible for an LLM to accurately simulate a complex $\mathcal{S}_\tau$ with a fixed context.

To overcome this, we develop a novel *context-adaptive simulation strategy* which adjusts the context supplied to the LLM mid-generation, based on the information requirements of the current simulation state. By continuously re-evaluating the relevance of the available information in $D_\tau$ and $K_\tau$ to the current simulation state we ensure that only the most useful context is given to the LLM, enabling it to simulate across large state-action spaces with varied dynamics without overloading its context window.

## 4. CALM-DT

We propose CALM-DT, a **C**ontext-**A**daptive **L**anguage **M**odel-based **D**igital **T**win (overview in Figure 2). CALM-DT is a simulation approach that allows an LLM to accurately model the dynamics of a system across diverse state-action pairs by continuously adjusting its context to minimise the generation error at each simulation state. A user inputs the target's state-action history ($h$), a dataset of related trajectories ($D_\tau$), a knowledge base including data and variable descriptions ($K_\tau$), an action policy ($\pi$), a context sample-set size ($c$), a rolling lookback ($l$), a resampling buffer ($r$), and a simulation horizon ($F$). CALM-DT simulates $F$ state-action pairs via an iterative three-stage process: **information retrieval** (§4.1), **prompt formulation** (§4.2), and **generation** (§4.3).

### 4.1. Information Retrieval

In this stage, relevant knowledge is extracted from $K_\tau$ and the top $c$ samples are retrieved from $D_\tau$, based on $h$.

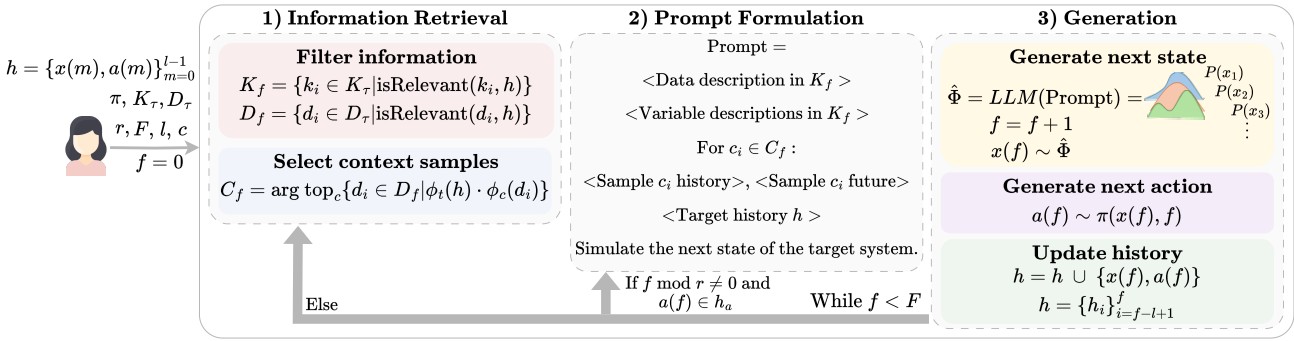

*Figure 2.* Simulation with CALM-DT. **Inputs:** target history ($h$), dataset of related trajectories ($D_\tau$), knowledge base including system and variable descriptions ($K_\tau$), action policy ($\pi$), context sample-set size ($c$), rolling lookback ($l$), resampling buffer ($r$), simulation horizon ($F$). For $F$ steps: **1)** Extract relevant knowledge from $K_\tau$ and choose the top $c$ samples from $D_\tau$ using fine-tuned encoders (§4.1). **2)** Construct natural language prompt describing the target and related systems (§4.2). **3)** Draw the next simulated state from the LLM, the next action from $\pi$, and update $h$ (§4.3). If a new action is taken, or $r$ steps have been simulated, return to **1)**, else return to **2)**.

### 4.1.1. KNOWLEDGE EXTRACTION

$K_\tau$ forms the basis for the knowledge-driven insights that will guide simulation. Through $K_\tau$ the user can encode their own knowledge into simulations, and assist the LLM to leverage its domain knowledge by providing it valuable context (Du et al., 2024). We consider a flexible $K_\tau$ with varying levels of information, depending on user preference. This can include general descriptions of the target system (e.g. *'This data describes a patient with cystic fibrosis'*), state variables (e.g. *'FEV1 is forced expiratory volume in 1 second'*) and action variables (e.g. *'Dornase alfa can stabilise or slow the decline of FEV1'*). Furthermore, any information that the user wishes to provide that represents their own priors over expected dynamics can also be included (e.g. *'FEV1 tends to decline without treatment'*). Crucially, this allows users to instil knowledge-based insights into simulation with *semantic descriptions*, not requiring distillation into well-formed equations, which can be a difficult task (Kacprzyk & van der Schaar, 2025). During knowledge extraction, any descriptions involving the state-action variables in $h$ are extracted from $K_\tau$, along with general system descriptions, to form the *relevant knowledge base* $K_f$.

### 4.1.2. SAMPLE RETRIEVAL

$D_\tau$, a dataset of $N$ state-action trajectories from related systems (e.g. other CF patients), forms the basis for the data-driven insights that will guide simulation. We wish to select only the most useful samples for the current $h$.

We firstly filter $D_\tau$ for samples with relevant actions, such that the DT can learn action-induced dynamics. Let $h_a = \text{unique}(\{a(t)\}_{t=0}^{l-1})$ denote the unique actions in $h$. We filter $D_\tau$ for samples $d_i$ whose action sequences $\{a_i(t)\}_{t=0}^{T_i-1}$ contain a subsequence of length at most $l$ that includes all actions in $h_a$, and that is followed by at least one future time step. Formally, the set of valid samples is $V = \{d_i \in$

$D_\tau \mid \exists\, t^*,\ w \le l:\ h_a \subseteq \{a_i(t)\}_{t=t^*}^{t^*+w-1}$, and $t^* + w \le T_i - 1\}$.[3] From each $d_i \in V$, we extract a *history-future* pair, where the history is a length $l$ state-action sequence covering the identified window that shares actions with $h$, and the future is the next (up to) $r$ state-action pairs. This forms the *action-relevant data base* $D_f$.

For sample selection, we adapt bi-encoder retrieval methods from natural language processing (NLP) (Cheng et al., 2023; Liu et al., 2024a) to our use case. Given we can convert state-action sequences into natural language representations via $g$, we can utilise natural language encoders $\phi_t$, $\phi_c$ to assess the similarity between a target and candidate sample. However, since time-series data differ significantly from typical NLP training distributions, we need to align pre-trained encoder embedding spaces with our task via fine-tuning. Furthermore, we enhance each trajectory's textual representation by appending an LLM-generated natural language summary of its trends. This enriched representation facilitates more effective retrieval by improving the encoder's ability to identify temporal patterns.

We employ contrastive learning (Becker & Hinton, 1992; Hadsell et al., 2006; Chen et al., 2020) to fine-tune $\phi_t$ and $\phi_c$, using LLM performance feedback to determine positive and negative samples. We construct a training dataset of target-candidate trajectory pairs, where multiple candidates are paired with each target. Each candidate is assigned a score based on the performance that an LLM achieves using it as a context sample when simulating the target trajectory.[4] For each target, we designate its paired candidate with the best score as the positive sample and its $B$ worst-scoring paired candidates as negative samples. We fine-tune $\phi_t$ and $\phi_c$ using the InfoNCE loss (Oord et al., 2018), to

---

[3]For continuous actions, appropriate discretisation is required.

[4]Scores can be metrics such as MSE, MAE, or Continuous Ranked Probability Score (CRPS) (Gneiting & Raftery, 2007).

maximise similarity between targets and their positive candidates, while minimising similarity to negative candidates. For each target $t$, and their respective positive ($c^+$) and negative ($\{c_i^-\}_{i=1}^B$) candidates, this loss is:

$$\mathcal{L} = -\log \frac{\exp\left(\frac{\phi_t(t) \cdot \phi_c(c^+)}{\tau}\right)}{\exp\left(\frac{\phi_t(t) \cdot \phi_c(c^+)}{\tau}\right) + \sum_{i=1}^B \exp\left(\frac{\phi_t(t) \cdot \phi_c(c_i^-)}{\tau}\right)} \tag{1}$$

where $\tau$ is a temperature parameter that controls the concentration of the distribution. This fine-tuning process encodes some notion of LLM simulation performance into the embedding spaces of $\phi_t$ and $\phi_c$, leading to retrieval of samples that enhance simulations for a given target. Notably, we have *not* contradicted our claim that CALM-DT can adapt to new modelling environments without parameter updates. We perform fine-tuning *only once*, in the initial modelling environment $\mathcal{E}_0$, using $D_0$ to form the training dataset, and the resulting encoders remain applicable across the proceeding environments, as their inputs are natural language trajectory representations that are dimension-agnostic. Moreover, as demonstrated in §6.2, even without fine-tuning we surpass existing DT approaches, indicating that a fully training-free version of CALM-DT is feasible (further details of this fine-tuning process, including elaboration on our LLM-generated time-series summaries, are in Appendix A).

At inference time, $\phi_t$ and $\phi_c$ are used to compute similarity scores between $h$ and each history of $d_i \in D_f$. The top $c$ highest-scoring samples are selected to form the context set $C_f = \text{top}_c\{d_i \in D_f | \phi_t(h) \cdot \phi_c(d_i)\}$.

### 4.2. Prompt Formulation

Once $K_f$ and $C_f$ have been determined, they are structured into a prompt to be supplied to the LLM. The general prompt format is shown in Figure 2, which includes each description $k_i \in K_f$, each history-future pair in $C_f$, and the target system's history. Example prompts are provided in Appendix B.

### 4.3. Generation

Supplied with this prompt, the LLM estimates the transition distribution of the target system, $\hat{\Phi}$, generating a distribution over the natural language representation of the next possible state. Depending on whether uncertainty quantification is desired, sampling can be performed using beam search (Sutskever et al., 2014), to obtain multiple plausible outcomes, or greedy decoding. For each state sampled from $\hat{\Phi}$, we apply the inverse mapping $g^{-1}$ to transform it into a structured state representation, and the corresponding action is then sampled from $\pi$. Generated state-action pairs are appended to $h$, and the oldest entry is removed to maintain a consistent rolling lookback $l$.

Finally, depending on the new $h$, the decision is made whether to return to **1) information retrieval**, or **2) prompt formulation**. If the newest action is not in the previous $h_a$, or the resampling buffer is exceeded (i.e., $f \mod r = 0$), the process returns to **1)** to re-select the most relevant information. Otherwise, the simulation continues from **2)** with the same information base.

To the best of our knowledge, CALM-DT is the first context-adaptive simulation method, dynamically adjusting its knowledge and data mid-generation. Notably, CALM-DT is LLM-agnostic, allowing us to leverage arbitrary base LLMs as $\hat{\Phi}$ without any necessary fine-tuning to incorporate relevant data- and knowledge-driven insights. By efficiently selecting and adapting the supplied context, CALM-DT permits accurate simulation across diverse state-action trajectories while maintaining manageable context window sizes. Additionally, novelty arises in our sample-selection method, as we are the first to propose retrieval of time-series data by leveraging LLM-generated summaries to enhance NLP encoder capabilities.

Crucially, CALM-DT allows continuous and seamless updating, as all data- and knowledge-driven insights arise purely from in-context learning. This ensures the model can easily adapt to new modelling environments without redesigns or parameter updates—an advantage that no existing DT approach offers.

## 5. Related Works

### 5.1. Digital Twins

DTs have been deployed as far back as the 1960s, where they were used by NASA to simulate spacecraft (Allen, 2021) based on a collection of expertly crafted equations derived from physical laws. While expert-defined DTs remain in use (Laubenbacher et al., 2024), they require significant effort on the part of domain experts to distil knowledge into mechanistic equations, which can limit scalability. Methods like genetic programming (Koza, 1994) and Sparse Identification of Nonlinear Dynamics (SINDy) (Brunton et al., 2016) aim to automate the discovery of such models, but these techniques are limited to relatively simple equations, and they result in fixed-dimensional models.

Deep learning approaches offer an alternative by approximating complex dynamics directly from longitudinal data. There are a variety of model architectures that are designed to handle sequential data, with the most powerful being those that can model long-term dependencies, including RNNs (Elman, 1990; Graves, 2012) and transformers (Wu et al., 2021; Zhou et al., 2022; Melnychuk et al., 2022). Neural ODEs (Chen et al., 2018; Dupont et al., 2019; Alvarez et al., 2020) extend deep learning capabilities to modelling continuous-time dynamics. Despite their expressiveness,

deep learning models require fixed input-output structures that are set at design time, obliging their re-design and re-training upon a change in modelling environment.

Hybrid approaches combine mechanistic equations with deep learning methods to improve sample efficiency and generalisation to out-of-distribution data. These range from models that combine neural components with well-defined physical equations (Raissi et al., 2019; Kuang et al., 2024) to those that integrate learnable components into known model structures (Qian et al., 2021; Takeishi & Kalousis, 2021). Recent efforts to automate DT creation leverage LLMs for automatic generation of hybrid DT code structures (Holt et al., 2024).

Despite these varied approaches, existing DT methods break-down upon changes in modelling environment, as they cannot easily incorporate additional variables or learn from new data or knowledge without extensive re-designs and re-training. Makarov et al. (2024) have recently considered direct use of LLMs as DTs, however they require LLM fine-tuning, hindering their ability to continuously learn as new data is released. To our knowledge we are the first to propose a DT method that can use *any* base LLM, without fine-tuning, uniquely allowing CALM-DT to adapt to changes in modelling environment.

In Table 1 we compare CALM-DT with a variety of existing DTs. Notably, CALM-DT allows the most general form knowledge inputs, allows uncertainty in simulations if desired, and is the only method to satisfy our desiderata.

### 5.2. World Models

World models are a related research area concerned with modelling environment dynamics to enable planning (Ha & Schmidhuber, 2018). While traditional world models typically focus on fixed-horizon, often discrete-time roll-outs within largely fixed environments, DTs can be seen as a specific class of world model, with some additional requirements. DTs must exhibit continuous-time dynamics, to allow simulation with arbitrary temporal granularity (Chen et al., 2025), and continuously update alongside their physical counterparts (Tzachor et al., 2022; National Academy of Engineering and National Academies of Sciences, Engineering, and Medicine, 2024). While typical world models focus on more static environments, DTs must be designed to update. Several recent works have explored using LLMs as world models (Hao et al., 2023; Liu et al., 2024c; Xie et al., 2025), however since they are not focused on applications as DTs specifically, they typically employ fixed prompting, or fine-tuning, and they do not explore updating the model upon changes in environment. Furthermore, using data from related systems for in-context learning to enhance LLM-based simulations is unexplored.

*Table 1.* Comparison of a collection of related DTs. **Knowledge**: How are knowledge-driven insights incorporated? **Probabilistic**: Allows uncertainty in simulation? **[D1]**: Handles arbitrary state-action dimensions without re-design? **[D2]**: Incorporates new information without parameter updates?

| Method | Knowledge | Probabilistic | [D1] | [D2] |
|---|---|---|---|---|
| Expert DTs | Equations | ✓ | ✗ | ✗ |
| Transformer | — | ✓ | ✗ | ✗ |
| RNN | — | ✓ | ✗ | ✗ |
| DyNODE | — | ✗ | ✗ | ✗ |
| SINDy | — | ✗ | ✗ | ✗ |
| HDTwin | Equations | ✗ | ✗ | ✗ |
| CALM-DT | Nat. language | ✓ | ✓ | ✓ |

### 5.3. LLM-Based Time-Series Forecasters

A related, yet distinct, area is time-series forecasting, where LLMs are gaining traction (Xue & Salim, 2023; Zhou et al., 2023; Jin et al., 2024; Gruver et al., 2024). While these methods show promising results, they focus solely on predicting future values and do not incorporate actions or allow for policy simulation, making them unsuitable for DT applications. Again, using data from related systems for in-context learning with LLM time-series forecasters is unaddressed.

## 6. Empirical Investigation

We now demonstrate the empirical performance of CALM-DT. Firstly, we examine simulations in *fixed* modelling environments, demonstrating state-of-the-art performance (§6.1). We also conduct ablation studies to assess the contribution of different components of CALM-DT (§6.2). We then showcase CALM-DT's unique ability to adapt to changes in modelling environment without re-design or re-training, demonstrating adaptation to a novel action (§6.3), and incorporation of new data (§6.4). We report detailed experimental set-ups in Appendix C.

### 6.1. Performance in Fixed Modelling Environments

**Setup.** We consider DT simulations in two medical scenarios: CF progression under treatment with dornase alfa (Yang & Montgomery, 2021), and non-small cell lung cancer (NSCLC) tumour growth under chemo- and radiotherapy. For the CF setting, we use 1000 trajectories from the 2008-2013 UK CF registry for training, and assess three-year simulation performance. For the NSCLC setting, we generate 500 training samples of 60-day cancer progression according to the pharmacokinetic-pharmacodynamic model from Geng et al. (2017), which has previous use in ML literature (Bica et al., 2020; Seedat et al.,

*Table 2.* DT simulation comparisons for CF and NSCLC progression. Averaged over 10 runs, with 95% CIs. Ordered by average ranking.

| | CF | | NSCLC | | |
| | MSE (↓) | MAE (↓) | MSE (↓) | MAE (↓) | Rank (↓) |
|---|---|---|---|---|---|
| 1-NN | $107 \pm 0$ | $6.96 \pm 0$ | $181.22 \pm 0$ | $6.12 \pm 0$ | 7.5 |
| RNN | $340 \pm 4.34$ | $13.8 \pm 0.090$ | $115 \pm 4.35$ | $6.04 \pm 0.645$ | 7.25 |
| Constant | $86.8 \pm 0$ | $5.84 \pm 0$ | $279.23 \pm 0$ | $7.21 \pm 0$ | 7 |
| SINDy | $73.0 \pm 0$ | $5.47 \pm 0$ | $1.90 \times 10^4 \pm 0$ | $37.6 \pm 0$ | 6.5 |
| Transformer | $83.1 \pm 1.01$ | $6.24 \pm 0.029$ | $116 \pm 52.7$ | $5.86 \pm 2.44$ | 6 |
| DyNODE | $82.6 \pm 0.279$ | $6.23 \pm 0.009$ | $104 \pm 63.6$ | $4.67 \pm 2.06$ | 4.5 |
| $K$-NN | $65.1 \pm 0$ | $5.47 \pm 0$ | $97.28 \pm 0$ | $4.91 \pm 0$ | 3 |
| HDTwin | $69.8 \pm 3.69$ | $5.44 \pm 0.169$ | $80.6 \pm 11.8$ | $3.42 \pm 0.717$ | 2 |
| CALM-DT | $\mathbf{55.3 \pm 0.811}$ | $\mathbf{4.63 \pm 0.045}$ | $79.4 \pm 8.57$ | $4.28 \pm 0.157$ | 1.25 |

2022; Melnychuk et al., 2022; Holt et al., 2024), and we assess 30-day simulation performance. Since the CF data is not publicly accessible, and we generate the NSCLC data from a stochastic simulator, the test data is unlikely to appear in any LLM training corpora, addressing potential data leakage issues.

In Table 2 we compare with three simple baselines: constant prediction, as well as $K$-nearest-neighbour predictions with $K = 1$ and with the best observed $K$. We also compare with five existing DT methods, using the implementations from Holt et al. (2024): an RNN (Graves et al., 2007), Transformer (Melnychuk et al., 2022), neural ODE model (DyNODE) (Alvarez et al., 2020), mechanistic model discovery method (SINDy) (Brunton et al., 2016), and LLM-driven hybrid DT discovery method (HDTwin) (Holt et al., 2024).

**Takeaway.** Assessing simulation performance in terms of MSE and MAE from the test sample futures, CALM-DT achieves the best ranking, averaged across both metrics and datasets. For CF simulation in particular, CALM-DT outperforms all other methods to a statistically significant degree. LLMs, using their inherent domain knowledge along with the context samples provided by our sample selection method, can generate accurate simulations of complex dynamical systems, with interventions, over horizons with both a small (CF) and large (NSCLC) number of time steps.

## 6.2. Ablation Studies

**Setup.** We conduct several ablation studies on CALM-DT in the CF setting from §6.1. In Table 3 we compare a variety of approaches to selecting the context set $C_f$. We consider (i) zero-shot generation ($C_f = \emptyset$), (ii) random selection, (iii) selection with encoders that are not fine-tuned, (iv) selection

with fine-tuned encoders without natural language time-series summaries appended to their inputs, (v) selection with encoders that are neither fine-tuned nor given summaries, (vi) selection by querying an LLM for the top $c$ samples, and (vii) placing the entire dataset in context ($C_f = D_f$).

On the left of Figure 3 we compare the performance of CALM-DT across different base LLMs, setting either GPT-4o, GPT-4o Mini, or GPT-3.5 Turbo as $\hat{\Phi}$. On the right we compare CALM-DT across different context set sizes $c \in \{0, 1, 2, 3, 5, 10\}$.

*Table 3.* CALM-DT CF performance with different $C_f$ selection methods. Averaged over 10 runs, with 95% CIs. Ordered by MSE.

| $C_f$ Selection Method | MSE (↓) | MAE (↓) |
|---|---|---|
| No fine-tuning/summary | $76.3 \pm 1.61$ | $5.15 \pm 0.0713$ |
| $C_f = \emptyset$ | $74.8 \pm 0.411$ | $5.24 \pm 0.021$ |
| Random | $73.8 \pm 3.22$ | $5.22 \pm 0.100$ |
| No fine-tuning | $67.1 \pm 0.894$ | $4.95 \pm 0.040$ |
| No summary | $66.6 \pm 0.387$ | $5.05 \pm 0.0333$ |
| LLM selection | $65.6 \pm 2.19$ | $4.88 \pm 0.116$ |
| $C_f = D_f$ | $65.3 \pm 3.90$ | $4.81 \pm 0.102$ |
| CALM-DT | $\mathbf{55.3 \pm 0.811}$ | $\mathbf{4.63 \pm 0.045}$ |

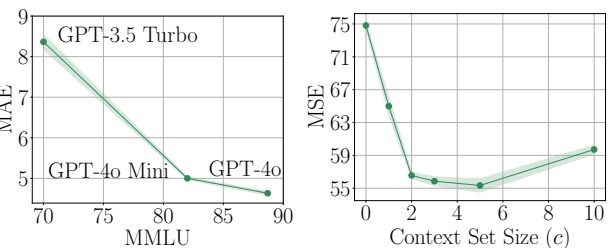

*Figure 3.* Left: Base LLM ablations. Right: Context set size ablations. Raw results in Appendix D

**Takeaway.**

1. The method of selecting $C_f$ significantly affects simulation performance. Setting $C_f = \emptyset$, conducting random selection, or using a non fine-tuned encoder without appended summaries all result in similar, relatively poor, performance. Adding either fine-tuning or appending natural language summaries improves encoder selection, and these approaches perform similarly to LLM-based selection and setting $C_f = D_f$. Our complete method, using a fine-tuned encoder with appended natural language summaries performs best, at a statistically significant level. It is interesting to note that the non fine-tuned encoder, with appended summaries, outperforms almost all DT approaches from Table 2, showing that an entirely training-free LLM-based DT can perform well.

2. Simulation performance scales with base LLM capacity (measured here with MMLU scores). Given CALM-DT is LLM-agnostic, this suggests that its performance will continue to improve along with ongoing LLM innovations, as opposed to existing DTs with fixed architectures.

3. Simulation performance rapidly improves with only a few context samples, showing how LLMs can effectively derive insights from small sample sets. $c = 5$ appears to be a point of saturation, with no further gains being realised for larger $c$. Conducting careful sample selection is therefore important, as filling the LLM context window with unnecessary samples hinders performance.

### 6.3. Introducing a New Action

**Setup.** We now demonstrate how CALM-DT naturally adapts to changes in modelling environment. In 2012, Ivacaftor was approved for CF treatment, and it is recorded in the UK CF registry from 2013. We investigate some changes in modelling environment that users could induce to include this new action, and how CALM-DT responds. We examine CALM-DT's performance when updating the:

1. *Action space only* ($\mathcal{A}$), where the new action is the only change from $\mathcal{E}_\tau \to \mathcal{E}_{\tau+1}$, and no information on the effects of Ivacaftor is given.

2. *Action space and knowledge base* ($\mathcal{A} + K$), where a high-level description of Ivacaftor's effects, derived from a 48-week clinical trial (Ramsey et al., 2011), is added to $K_\tau \to K_{\tau+1}$.

3. *Action space, knowledge base, and dataset* ($\mathcal{A} + K + D$), where samples from 2013 with one post-Ivacaftor measurement are also added to $D_\tau \to D_{\tau+1}$.

We report three-year simulation performance on patients that receive Ivacaftor from 2013-2015 in Table 4.

*Table 4.* CALM-DT simulation post-Ivacaftor across three adaptation scenarios. Averaged over 20 runs, with 95% CIs.

| Updates to $\mathcal{E}_\tau$ | MSE ($\downarrow$) | MAE ($\downarrow$) |
|---|---|---|
| $\mathcal{A}$ | $59.7 \pm 0.472$ | $4.95 \pm 0.0166$ |
| $\mathcal{A} + K$ | $54.4 \pm 0.472$ | $4.75 \pm 0.0242$ |
| $\mathcal{A} + K + D$ | $\mathbf{50.4 \pm 0.477}$ | $\mathbf{4.28 \pm 0.0251}$ |

**Takeaway.** Updating the modelling environment by adding the Ivacaftor action without providing any relevant information on its effects naturally leads to poor simulation. Expanding $K_\tau$ to include a high-level description of its effects greatly improves performance, and adding patients treated with Ivacaftor to $D_\tau$ leads to a further increase. LLMs can incorporate insights from natural language descriptions into simulations, avoiding the need to distil knowledge into well-defined equations, and they can effectively harmonise this with data-driven insights to maximise performance.

### 6.4. Incorporating New Data

**Setup.** We now show how CALM-DT can continuously learn over time by incorporating growing data into $D_\tau$. In §6.3 we only used data points from 2013 to expand $D_\tau$ with post-Ivacaftor information, showcasing performance immediately after a change in modelling environment. Longer term trajectories become available with each yearly release from the UK CF registry, and CALM-DT can easily incorporate this new data to improve its accuracy. To demonstrate this, we report performance in Table 5 with varying amounts of post-Ivacaftor data in $D_\tau$, from one to three years.

*Table 5.* CALM-DT simulation post-Ivacaftor with growing $D_\tau$. Averaged over 20 runs, with 95% CIs.

| Post-Ivacaftor Data | MSE ($\downarrow$) | MAE ($\downarrow$) |
|---|---|---|
| One year | $50.4 \pm 0.477$ | $4.28 \pm 0.0251$ |
| Two years | $50.0 \pm 1.00$ | $4.29 \pm 0.0468$ |
| Three years | $\mathbf{47.8 \pm 1.01}$ | $\mathbf{4.19 \pm 0.0523}$ |

> **Takeaway.** As more post-Ivacaftor data becomes available, CALM-DT learns its effects better, improving in three-year simulation accuracy. CALM-DT can effectively utilise growing datasets to improve performance without re-training.

### 6.4.1. COMPARISON WITH A DOMAIN-SPECIFIC MODEL

For further context, we compare with a recent expert-derived linear mixed-effects model for CF progression under Ivacaftor. Zhou et al. (2024) model CF patients' `FEV1PP`—forced expiratory volume in 1 second as a percentage of expected performance, a critical value for measuring CF progression—using data from the US CF foundation patient registry (Knapp et al., 2016). They report an RMSE of 6.78 for six-month predictions on a validation cohort. `FEV1PP` is one of the state variables we model in our CF setting, so we can compare to this. Using three years of post-Ivacaftor data in $D_\tau$, CALM-DT achieves an RMSE for `FEV1PP`, on our testing cohort, of 8.11 over one year, and 8.58 over three years.[5] Using Zhou et al. (2024) as a domain-specific upper bound on performance, we see that CALM-DT performs relatively well, even with its low expertise requirements and seamless adaptability to changes in environment.

## 7. Discussion

Our empirical results demonstrate that CALM-DT generates state-of-the-art simulations in fixed modelling environments (§6.1). We validate our specific design choices in CALM-DT (§6.2), and show how it can model novel actions (§6.3) and incorporate new data to guide its simulations (§6.4). Unlike existing DT approaches, which require architectural modifications and parameter updates to extend state-action spaces and incorporate new information, CALM-DT can adapt without re-training, since it relies on in-context learning alone, enabling effective use in real-world, dynamic environments.

### 7.1. Limitations

There are, of course, limitations to CALM-DT. Inference speed will generally be slower than other DT approaches, especially over extended simulation horizons where the encoders and base LLM are called multiple times. Furthermore, if using closed-sourced LLMs via API calls, monetary costs must be considered.

There are also specific artefacts of LLM-based simulations that users should be aware of. It is well known that LLMs can hallucinate (Maynez et al., 2020; Ji et al., 2023), which

could result in unlikely spikes or dips being included in simulations, or generation of values that violate known domain constraints. Hallucinations can be difficult to predict or detect automatically, so careful analysis of simulation outputs is critical. Biases in LLM training corpora can influence model outputs (Bender et al., 2021), potentially leading to disparities in simulation performance across populations. Classical tokenisation schemes (e.g. byte-pair encoding (Sennrich et al., 2016)) can limit LLM performance in numerical tasks (Liu & Low, 2023; Gruver et al., 2024), obliging caution in scenarios where high precision is necessary. Modern tokenisation schemes, however, are improving numerical handling (Touvron et al., 2023). Finally, correctly structured outputs cannot be guaranteed, given the stochastic nature of LLMs, although, in practice, we rarely experienced such issues. At times, textual explanations or Markdown characters were erroneously included in simulation outputs. The majority of these limitations will likely benefit from general innovations in LLM technology.

### 7.2. Future Works

Potential future directions for CALM-DT include extensions to improve generalisation and adaptability. Generalisation could be improved by incorporating causal graphs into simulation, to ensure known causal relationships are respected, and expanding our information retrieval step to utilise large, unstructured textual corpora with potentially extensive insights. To improve adaptability, focus can be placed on actively retrieving information to reduce uncertainty during deployment (Kobalczyk et al., 2025), including strategic selection of informative data samples, which may be costly to acquire (Astorga et al., 2024).

## Impact Statement

This paper presents work whose goal is to advance the field of Machine Learning. There are many potential societal consequences of our work, none which we feel must be specifically highlighted here.

## Acknowledgements

Harry Amad's studentship is funded by Canon Medical Systems Corporation.

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

## A. Bi-Encoder Fine-Tuning with LLM Feedback

We include extended details on our bi-encoder retrieval fine-tuning process here.

**Time-Series Natural Language Representation.** Firstly, we define the mapping $g$ we use to produce a natural language representation of a time-series. Consider the following time-series with state variables $x$ and $y$, and action variable $z$.

$$[\{x:1, y:1, z:0\}, \{x:2, y:1, z:0\}, \{x:3, y:1, z:1\}]$$

Where the measurements are taken at times $t = 0, 1, 2$. The natural language representation of the states of this series, as produced by our mapping $g$, is:

```
"""
Time 0: x: 1, y: 1 | Time 1: x: 2, y: 1 | Time 2: x: 3, y: 1
"""
```

And for the actions, it is:

```
"""
Time 0: z: 0 | Time 1: z: 0 | Time 2: z: 1
"""
```

**Training Dataset Construction.** Given a dataset of trajectories $D = \{\{x_i(t), a_i(t)\}_{t=1}^{T_i-1}\}_{i=1}^{N}$, we first construct a contrastive learning training dataset, in a manner similar to existing NLP approaches (Cheng et al., 2023; Liu et al., 2024a). Each sample in this dataset contains a target trajectory and $C$ candidate trajectories. We then score each candidate with a performance metric, such as MSE, MAE, or CRPS (Gneiting & Raftery, 2007), that is based on an LLM simulating the future of the target trajectory, using the candidate as an in-context learning example. Based on these scores, we designate candidates as either positive or negative samples. Specifically, the candidate that results in the best score is designated as the positive sample, and the $B$ worst performing candidates are the negative samples. Finally, for each target, and its respective positive and negative candidates, we generate a natural language summary of the trends in its state trajectory, using an LLM. This is to allow the encoders to more easily pick up on temporal trends, that may be difficult to assess from numerical values alone. We generate this trajectory summary by supplying an LLM with the following prompt:

```
"""
For each variable in this time-series, write <VARIABLE NAME>: <TREND>, where <TREND> is a
    list of one or more descriptive words that summarises the series in chunks. Decide how
     to chunk each variable based on when its trend changes. Neighbouring chunks should
    not have the same description. Each <TREND> each word is either [increasing,
    decreasing, stable]. There should be fewer chunks than points in the time-series. Time-
    series: {trajectory_str}
"""
```

This summary is then appended to the natural language representation of the trajectory as produced by the mapping $g$.

**Training.** With the training dataset constructed, we initialise two encoders $\phi_t$ and $\phi_c$ to encode the target and candidate samples, respectively. Then, for each target $t$, we extract their positive ($c^+$) and negative ($\{c_i^-\}_{i=1}^{B}$) candidates, and train the encoders with the InfoNCE loss (Oord et al., 2018):

$$\mathcal{L} = -\log \frac{\exp(\frac{\phi_t(t) \cdot \phi_c(c^+)}{\tau})}{\exp(\frac{\phi_t(t) \cdot \phi_c(c^+)}{\tau}) + \sum_{i=1}^{B} \exp(\frac{\phi_t(t) \cdot \phi_c(c_i^-)}{\tau})} \tag{2}$$

where $\tau$ is a temperature parameter that controls the concentration of the distribution.

## B. Example Prompts

Here are example prompts input to the LLM for simulation of a CF and NSCLC patient, produced by the **Prompt Formulation** step of CALM-DT as described in §4.

**B.1. CF Example Prompt**

```
"""
The data is from a patient with cystic fibrosis. The time unit is in years.

STATE VARIABLES:
FEV1PP: Forced expiratory volume in 1 second compared to the standard for that age.
Weight: Patient weight in kg.
Height: Patient height in cm.

ACTION VARIABLES:
Dornase_Alfa: A treatment which can stabilise or slow the decline of FEV1.

Example 1 state history:
Time 2008: FEV1PP: 80.1, Weight: 65.0, Height: 168 | Time 2009: FEV1PP: 81.0, Weight:
    66.2, Height: 168 | Time 2010: FEV1PP: 74.2, Weight: 63.5, Height: 168

Example 1 action history:
Time 2008: Dornase_Alfa | Time 2009: Dornase_Alfa | Time 2010: Dornase_Alfa

Example 1 state future:
Time 2011: FEV1PP: 71.2, Weight: 64.1, Height: 168

Given the following state history:
Time 2008: FEV1PP: 77.8, Weight: 70.1, Height: 174 | Time 2009: FEV1PP: 80.1, Weight:
    69.8, Height: 174 | Time 2010: FEV1PP: 74.0, Weight: 70.9, Height: 174

And the following action history:
Time 2008: Dornase_Alfa | Time 2009: Dornase_Alfa | Time 2010: Dornase_Alfa

Simulate the next timestep's state, for all state variables. Follow the exact format of
    the state history.
"""
```

**B.2. NSCLC Example Prompt**

```
"""
The data describes treatment responses for combined chemo and radiation therapy for non-
    small cell lung cancer patients, generated from a bio-mathematical model. The time
    unit is in days.

STATE VARIABLES:
tumour_volume: Volume of the tumour with units cm^3.
chemotherapy_drug_concentration: Concentration of the chemotherapy drug vinblastine with
    units mg/m^3.

ACTION VARIABLES:
chemotherapy_dosage: Dosage of the chemotherapy drug vinblastine with units mg/m^3.
radiotherapy_dosage: Dosage of the radiotherapy with units Gy.

Example 1 state history:
Time 0: tumour_volume: 734.27, chemotherapy_drug_concentration: 0 | Time 1: tumour_volume:
     200.94, chemotherapy_drug_concentration: 0 | Time 2: tumour_volume: 227.94,
    chemotherapy_drug_concentration: 0 | Time 3: tumour_volume: 248.87,
    chemotherapy_drug_concentration: 0 | Time 4: tumour_volume: 161.67,
    chemotherapy_drug_concentration: 5.01 | Time 5: tumour_volume: 118.61,
    chemotherapy_drug_concentration: 4.06 | ... | Time 29: tumour_volume: 4.82,
    chemotherapy_drug_concentration: 11.26

Example 1 action history:
Time 0: chemotherapy_dosage: 0, radiotherapy_dosage: 2 | Time 1: chemotherapy_dosage: 0,
    radiotherapy_dosage: 2 | Time 2: chemotherapy_dosage: 0, radiotherapy_dosage: 0 | Time
     3: chemotherapy_dosage: 5, radiotherapy_dosage: 0 | Time 4: chemotherapy_dosage: 0,
```

```
    radiotherapy_dosage: 0 | Time 5: chemotherapy_dosage: 0, radiotherapy_dosage: 0 | ...
    | Time 29: chemotherapy_dosage: 0, radiotherapy_dosage: 0

Example 1 state future:
Time 30: tumour_volume: 4.52, chemotherapy_drug_concentration: 5.62

...

Given the following state history:
Time 0: tumour_volume: 429.03, chemotherapy_drug_concentration: 0 | Time 1: tumour_volume:
    625.94, chemotherapy_drug_concentration: 5.00 | Time 2: tumour_volume: 602.07,
    chemotherapy_drug_concentration: 9.32 | Time 3: tumour_volume: 539.74,
    chemotherapy_drug_concentration: 10.36 | Time 4: tumour_volume: 385.34,
    chemotherapy_drug_concentration: 5.04 | Time 5: tumour_volume: 270.77,
    chemotherapy_drug_concentration: 6.30 | ... | Time 29: tumour_volume: 11.19,
    chemotherapy_drug_concentration: 8.24

And the following action history:
Time 0: chemotherapy_dosage: 5, radiotherapy_dosage: 0 | Time 1: chemotherapy_dosage: 5,
    radiotherapy_dosage: 0 | Time 2: chemotherapy_dosage: 5, radiotherapy_dosage: 2 | Time
    3: chemotherapy_dosage: 5, radiotherapy_dosage: 2 | Time 4: chemotherapy_dosage: 5,
    radiotherapy_dosage: 2 | Time 5: chemotherapy_dosage: 0, radiotherapy_dosage: 0 | ...
    | Time 29: chemotherapy_dosage: 0, radiotherapy_dosage: 0

Simulate the next timestep's state, for all state variables. Follow the exact format of
    the state history.
"""
```

### B.3. System Prompt

Additionally, we provide the LLM with the following system prompt to encourage the output to follow proper formatting:

```
"""
You are an expert at simulating dynamical systems. Respond only with the simulation in the
    exact format requested. Do not use the characters * or - anywhere. Ensure that you
    simulate exactly the desired number of timesteps for each state variable.
"""
```

## C. Experiment Details

We now detail our set-ups for the experiments in §6.

### C.1. Fixed CF Environment

For this experiment, we use data from the UK Cystic Fibrosis Registry[6] from 2008-2013, which records annual follow-ups for patients with CF in the UK. We extract the first 1000 patients that have annual recordings in every year from 2008-2013 (ordered by patient ID), and this acts as the training dataset for each method. We extract the next 100 patients, and use this as a validation set. Finally, we extract the next 50 patients, and use this as the testing set. For each patient, we consider three critical state variables:

FEV1PP: The forced expiratory volume in 1 second as a percentage compared to the standard for people of that age. This is a critical value for measuring CF progression, where low FEV1PP scores indicate advanced disease progression.

Weight: Patient weight in kg. Advanced CF progression can lead to weight lose due to malabsorption of nutrients and increased energy expenditure.

Height: Patient height in cm. Advanced CF progression can lead to growth impairment due to malabsorption of nutrients and potential loss of bone density.

---

[6] https://www.cysticfibrosis.org.uk/

We also consider the action variable:

`Dornase_Alfa`: A binary action, denoting if the patient is receiving treatment with dornase alfa. Dornase alfa is a treatment used to manage CF, which can assist with mucus breakdown in CF patients, reducing viscosity and improving airway clearance. It helps lung function and reduces the frequency of respiratory infections, potentially stabilising or slowing the decline of `FEV1PP`.

We ensure that each patient has annual recordings for each of the above variables from 2008-2013. For the testing set, we supply each method with three-year input history, from 2008-2011, and assess state simulation error for the next three years (2011-2013).

For CALM-DT, we use GPT-4o, accessed via the Azure OpenAI Service with version `2024-02-01`, as the base LLM with the temperature $\tau = 0$, and we set $K_\tau$ as:

```
{
'General description': 'The data is from a patient with cystic fibrosis. The time unit is
    in years.'
'FEV1PP': 'Forced expiratory volume in 1 second compared to the standard for that age.'
'Weight': 'Patient weight in kg.'
'Height': 'Patient height in cm.'
'Dornase_Alfa: 'A treatment which can stabilise or slow the decline of FEV1.'
}
```

For $D_\tau$ we use the training dataset of 1000 patients. We also set $r = 1$, $l = 3$, $F = 3$, $c = 5$.

We conduct bi-encoder retrieval fine-tuning as described in Appendix A using $D_\tau$ to construct the contrastive learning training dataset. For each sample, we choose $C = 5$ candidate samples, and we set the $B = 2$ lowest scoring as the negative candidates for each target sample. We use GPT-4o Mini to generate textual summaries of the time-series for this training dataset. We score the candidate samples using the CRPS calculated from five simulated three-year futures compared to the true future, using GPT-4o Mini for simulation. We use `ModernBERT` (Warner et al., 2024) as the base for the encoders $\phi_t$ and $\phi_c$, using the implementation from the `Transformers` Python library (Wolf et al., 2020). We conduct training for 8 epochs with a batch size of 16, learning rate of $5 \times 10^{-5}$, and temperature of $\tau = 0.07$, using the AdamW optimizer (Kingma & Ba, 2015) as implemented in PyTorch (Paszke et al., 2019).

### C.2. Fixed NSCLC Environment

For this experiment, we generate data from a bio-mathematical pharmacokinetic- pharmacodynamic model designed in Geng et al. (2017), which models tumour volume under chemo- and radiotherapy. We generate trajectories for 500 patients with 60 daily time-steps each. We generate validation and testing sets of 100 patients each. Each patient trajectory contains two state variables:

`tumour_volume`: Cancer tumour volume, in cm$^3$. This is the primary measurement for NSCLC disease progression.

`chemotherapy_drug_concentration`: Concentration of the chemotherapy drug vinblastine in the patient, in mg/m$^3$.

We also consider two action variables:

`chemotherapy_dosage`: Current dosage of the chemotherapy drug vinblastine, in mg/m$^3$.

`radiotherapy_dosage`: Radiation dosage, in Gy.

In testing, we supply each method with an input 30-day history and assess simulation error for the next 30 days.

For CALM-DT, we use GPT-4o, accessed via the Azure OpenAI Service with version `2024-02-01`, as the base LLM with the temperature $\tau = 0$, and we set $K_\tau$ as:

```
{
'General description': 'The data describes treatment responses for combined chemo and
    radiation therapy for non-small cell lung cancer patients, generated from a bio-
    mathematical model. The time unit is in days.'
'tumour_volume': 'Volume of the tumour with units cm^3.'
'chemotherapy_drug_concentration': 'Concentration of the chemotherapy drug vinblastine
    with units mg/m^3.'
```

```
'chemotherapy_dosage': 'Dosage of the chemotherapy drug vinblastine with units mg/m^3.'
'radiotherapy_dosage: 'Dosage of the radiotherapy with units Gy.'
}
```

For $D_\tau$ we use the training dataset of 500 patients. We also set $r = 1$, $l = 30$, $F = 30$, $c = 5$.

We conduct bi-encoder retrieval fine-tuning as described in Appendix A using $D_\tau$ to construct the contrastive learning training dataset. For each sample, we choose $C = 3$ candidate samples, and we set the $B = 2$ lowest scoring as the negative candidates for each target sample. We use GPT-4o to generate textual summaries of the time-series for this training dataset. We score the candidate samples with the average MSE between three simulated 10-day futures and the true future, using GPT-4o for simulation. We use `ModernBERT` (Warner et al., 2024) as the base for the encoders $\phi_t$ and $\phi_c$, using the implementation from the `Transformers` Python library (Wolf et al., 2020). We conduct training for 3 epochs with a batch size of 16, learning rate of $5 \times 10^{-5}$, and temperature of $\tau = 0.07$, using the AdamW optimiser (Kingma & Ba, 2015) as implemented in PyTorch (Paszke et al., 2019).

### C.3. Ablations

For the ablation experiments in §6.2, we use the standard CF dataset and set-up, except for the stated ablation change.

#### C.3.1. SAMPLE SELECTION

For the sample selection ablation, for the zero-shot method we set $c = 0$. For the random sampling method, instead of selecting using bi-encoder retrieval, we simply randomly sample $c = 5$ samples from $D_f$. For the encoder selection without fine-tuning, we set $\phi_t$ and $\phi_c$ as `ModernBERT` (Warner et al., 2024) and do not conduct fine-tuning. For the encoder selection without appended summaries, we do not append natural language summaries to encoder inputs during training and inference.

For the LLM-based selection, we construct the following prompt that instructs the LLM to choose the top $c$ each CF samples for the specific target:

```
"""
Target system history: <TARGET>.

Here are <N> related systems. Return only the indices of the <c> most similar histories to
    the target, with no other text. Do not repeat any indices. Separate the indices with
    commas

Related system 0 history: <DATASET[0]>
Related system 1 history: <DATASET[1]>
Related system 2 history: <DATASET[2]>
.
.
.
Related system <N> history: <DATASET[N]>

Indices of the <c> most similar histories:
"""
```

We use GPT-4o to conduct selection, using this prompt. Finally, for the full context ablation $C_f = D_f$, we supply all samples in $D_f$ as context. Note that the LLM-based selection and full context approaches are only applicable if the entire dataset can fit in the context window of an LLM. For the CF data, the entire dataset consists of approximately $100,000$ tokens, so it does fits into most modern LLM context windows, however this will generally not be the case for larger datasets.

#### C.3.2. BASE LLM

For the base LLM ablation, we set the base LLM as either GPT-4o (version `2024-02-01`), GPT-4o mini (version `2024-10-01-preview`), or GPT-3.5 Turbo (version `2024-10-01-preview`), all accessed via the Azure OpenAI service. We source the MMLU scores used in our visualisation from `https://paperswithcode.com/sota/multi-task-language-understanding-on-mmlu`.

### C.3.3. CONTEXT SET SIZE

For our context set size ablation, we vary $c \in \{0, 1, 2, 3, 5, 10\}$.

## C.4. Ivacaftor Experiments

### C.4.1. INTRODUCING A NEW ACTION

For the illustrative experiments used to demonstrate how CALM-DT can seamlessly adapt to changes in environment, we extract a different $D_\tau$ and test on a different testing set from the UK Cystic Fibrosis Registry. In this case, we extract all patient records from 2010-2013 that are treated with Ivacaftor in 2013 (its first year in the dataset) to use as $D_\tau$. This results in 168 patients, and we use the last 50 (ordered by patient ID) as the testing set (extending their records to 2010-2015, so we can test the three-year simulation accuracy with a three-year input history). We extract the same state variables as the previous experiment, however the action variable we consider now is `Ivacaftor`, which is a binary variable indicating whether the patient is receiving treatment with Ivacaftor.

Since we wish to use these experiments to investigate how CALM-DT would respond to a totally novel action, we want to construct a scenario where the base LLMs will have no internal knowledge about the action's effect, so that all its insights come from either the updated $K_\tau$ or $D_\tau$. Since Ivacaftor has well-known effects now, as it was introduced in 2012, we rename it in the dataset to `Drug X`, to remove the effect of any prior knowledge that the base LLMs may have about its efficacy for CF treatment. We also tried this experiment using a more realistic, yet still fake, CF drug name—`Pulmurex`—which is less clearly a placeholder than `Drug X`, to test whether the LLM could be speculating about what `Drug X` actually is, and if it affects the results. We noticed no difference between these settings.

We use the same fine-tuned encoders $\phi_t$ and $\phi_c$ for this experiment as in the previous experiment, showing that the encoders do not need re-training after environment changes, only requiring one initial fine-tuning run in environment $\mathcal{E}_0$. We use largely the same experimental settings as in §6.1, although now with the expanded $D_\tau$, and we add the following entry to $K_\tau$:

```
'Drug X': 'A cystic fibrosis treatment that clinical trial results suggest can initially
    improve lung function by 10.6 percentage points'
```

This description reflects a likely high-level takeaway that a user would have from reading the 48-week clinical trial on Ivacaftor prior to its approval in 2012 (Ramsey et al., 2011), without requiring excessive expertise nor effort to derive a more formal description.

### C.4.2. INCORPORATING NEW DATA

For §6.4 we use a similar set-up. However, we now consider how three-year simulation accuracy for patients with Ivacaftor would progress with each passing year, and the resulting annual release of data from the UK CF registry. For the setting 'One Year' we use data from 2010-2013 to form $D_\tau$, for 'Two Years' we use data from 2010-2014, and for 'Three Years' we use 2010-2015. With each extra year of data, the selected context samples from $D_\tau$ will have an extra year post Ivacaftor treatment in their respective futures.

## D. Raw Results for Ablation Plots

Here we provide the raw results used to create the ablation plots in §6.2. Table 6 contains simulation results across different base LLMs, while Table 7 contains simulation results across different context set sizes.

*Table 6.* Base LLM ablation results. Averaged over 10 runs, with 95% CIs.

| Base LLM | MSE ($\downarrow$) | MAE ($\downarrow$) |
|---|---|---|
| GPT-3.5 Turbo | $162.203 \pm 5.367$ | $8.364 \pm 0.183$ |
| GPT-4o Mini | $64.179 \pm 0.637$ | $5.001 \pm 0.032$ |
| GPT-4o | $55.336 \pm 0.811$ | $4.634 \pm 0.045$ |

*Table 7.* Context set size ablation results. Averaged over 10 runs, with 95% CIs.

| $c$ | MSE ($\downarrow$) | MAE ($\downarrow$) |
|---|---|---|
| 0 | $74.816 \pm 0.411$ | $5.241 \pm 0.021$ |
| 1 | $64.985 \pm 1.047$ | $5.013 \pm 0.053$ |
| 2 | $56.556 \pm 0.364$ | $4.681 \pm 0.010$ |
| 3 | $55.860 \pm 0.544$ | $4.674 \pm 0.029$ |
| 5 | $55.336 \pm 0.811$ | $4.634 \pm 0.045$ |
| 10 | $59.717 \pm 0.529$ | $4.891 \pm 0.024$ |

## E. Benchmark Methods

Here we detail the benchmark DT implementations. We implement three simple baselines of constant prediction (**Constant**), one-nearest-neighbour (**1-NN**), and $K$-nearest-neighbour (**K-NN**). For the more sophisticated DT baselines, we adapt the open source code from Holt et al. (2024), from the public GitHub https://github.com/samholt/HDTwinGen, for use with our target datasets, and we use largely the same hyperparameter settings as in their work. We compare against a neural ODE (Chen et al., 2018) (**DyNODE**) (Alvarez et al., 2020), a mechanistic discovery model (**SINDy**) (Brunton et al., 2016), a Transformer model (**Transformer**) (Melnychuk et al., 2022), an RNN (**RNN**), and an LLM-driven hybrid digital twin discovery method (**HDTwin**) (Holt et al., 2024).

**Constant**

For constant prediction, we simply repeat the last observed state in the input history for all steps in the simulation horizon.

**1-NN**

For 1-NN, we calculate the Euclidean distance between the target system's state history, and the state histories of all training examples. We select the most similar training sample, and use its future states as predictions for the target system.

**K-NN**

For $K$-NN, we calculate the Euclidean distance between the state history of the target system, and the state histories of all training examples. We select the $K$ most similar training samples, and use a weighted average of their futures (weighted by similarity) to predict the future of the target system. We report results for the the best performing $K$ value. For the CF dataset, we use $K = 12$. For NSCLC data, we use $K = 13$.

**DyNODE**

DyNODE is a neural network-based dynamics model (Alvarez et al., 2020), that models system dynamics by incorporating control into the standard neural ODE framework (Chen et al., 2018). We implement DyNODE with a 3-layer MLP, with a hidden dimension of 128, with tanh activation functions, and Xavier weight initialisation (Kumar, 2017). We optimise for the MSE of next-state prediction, using an Adam optimiser (Kingma & Ba, 2015), with a learning rate of 0.01, batch size of 1,000 and early stopping with a patience of 20 for 2,000 epochs.

**Transformer**

Causal Transformer is a state-of-the-art model for estimating counterfactual outcomes (Melnychuk et al., 2022). Due to the design for the estimation of counterfactual outcomes in treatment effect settings, this method typically incorporates three separate transformer networks, for processing covariates, past treatments, and past outcomes, respectively. We follow the implementation by Holt et al. (2024), and instead implement only a single transformer to model the past outcomes, which is applicable to our datasets. This consists of a standard transformer encoder, with input normalisation according to the training dataset. We encode input observed dimension of the state-action into an embedding vector dimension of size 250 through a linear layer, followed by the addition of a standard positional encoder (Melnychuk et al., 2022); this is then fed into a transformer encoder layer, with a head size of 10, dropout 0.1, and the output of this is then fed into a linear layer to reconstruct the next state. We train this model using the AdamW optimiser with a learning rate of 0.00005 and a step learning rate scheduler of step size 1.0 and gamma 0.95; we also implement gradient clipping to 0.7, with a batch size of 1,000 and early stopping with a patience of 20 for 2,000 epochs.

**RNN**

Recurrent Neural Networks (Graves et al., 2007) are widely used for time series prediction. We implement this with input normalisation according to the training dataset. The model consists of a gated recurrent unit RNN mapping the state-action dimension to a hidden dimension of size 250, with two layers. The output is then fed to a linear layer to convert the hidden dimension back to the state dimension for next step prediction. We use an Adam optimiser, with a learning rate of 0.01, batch size of 1,000 and early stopping with a patience of 20 for 2,000 epochs.

**SINDy**

Sparse Identification of Nonlinear Dynamics (SINDy) (Brunton et al., 2016) is a data-driven discovery method for the governing equations of a dynamical system. It iteratively performs sparse regression on a library of candidate functions to identify the best representation of the dynamical system. In our implementation, we use a polynomial library of order two. Finite difference approximations are used to compute time derivatives from the input time-series data, of order one. The alpha parameter is set at 0.5 and the sparsity threshold is set at 0.02.

**HDTwin**

HDTwin is an automatic hybrid DT discovery method, using a code-generating LLM to propose and iteratively refine DT model definitions involving white- and black-box components. The LLM proposes initially white-box mechanistic methods, which are fit to the training data and evaluated on a validation set. The evaluation metrics and then fed back into the LLM, and targeted improvements are made to the model, such as incorporating my complex, black-box components.

We use GPT-4o with temperature 0.7 as the code-generating LLM, allowing 20 generation/refinement iterations, with a patience of 5, and we select the best performing model on the validation set as the final DT definition. Each generation, the suggested model that is trained to optimise for the MSE of next state prediction, using the AdamW optimiser with a learning rate of 0.01 a batch size of 1,000, early stopping with a patience of 20 for 2,000 epochs. For the model refinement stage, we set $K = 16$ for the number of previous models to keep in memory and guide suggestions.

## F. Further Fixed Environment Results

We now report some further comparative results between CALM-DT and other DT methods in more fixed environments. We examine some simple di- and tritrophic ecological environments, of hare-lynx and algae-flagellate-rotifer population dynamics respectively, using the datasets from Bonnaffé & Coulson (2023). With these data, we wish to investigate how methods perform in scenarios with very small training sets, simple dynamics, and no actions. In these settings, traditional deep learning approaches can sometimes struggle due to over-fitting, failing to capture the underlying simple dynamics, while simpler baselines often perform best. For these experiments, due to the small datasets, encoder fine-tuning is impractical, so we use simple random selection to construct $C_f$ for CALM-DT, with a context set size $c = 2$ for the Hare-Lynx (HL) dataset and $c = 5$ for the Plankton dataset.

**Setup.**   We split the Hare-Lynx dataset into nine samples of 10 years each, and we set the first six samples as the training set, and use last three samples as the testing set, examining five-year simulation performance with a five-year input history. There are two state variables:

`Hare`: Annual count of hare pelts, serving as a proxy for the hare population size, in tens of thousands.

`Lynx`: Annual count of lynx pelts, serving as a proxy for the lynx population size, in tens of thousands.

We split the Algae-Flagellate-Rotifer dataset into 10 samples of 10 days each, and we set the first six samples as the training set, and use last four samples as the testing set, examining five-day simulation performance with a five-day input history. There are three state variables:

`algae`: Daily count of algae, serving as the primary prey.

`flagellate`: Daily count of flagellate, acting as the intermediate predators and prey.

`rotifer`: Daily count of rotifers, representing the top predators.

Simulation performances for the hare-lynx and algae-flagellate-rotifer data are reported in Tables 8 and 9, respectively.

**Takeaway.**   CALM-DT demonstrates its robust performance with this extended set of results. It achieves the best results among deep learning methods in these simpler, small-data environments. This indicates that CALM-DT, leveraging the

*Table 8.* DT simulation comparisons of ditrophic hare-lynx population dynamics. Sorted by MSE. Averaged over 20 runs, presented with 95% CIs.

| Model | MSE ($\downarrow$) | MAE ($\downarrow$) |
|---|---|---|
| HDTwin | $1.11 \times 10^4 \pm 1.93 \times 10^4$ | $29.6 \pm 9.21$ |
| Transformer | $2.52 \times 10^3 \pm 796$ | $31.7 \pm 5.44$ |
| Constant | $1.73 \times 10^3 \pm 0$ | $32.9 \pm 0$ |
| SINDy | $1.05 \times 10^3 \pm 0.00$ | $26.5 \pm 0.00$ |
| DyNODE | $895 \pm 212$ | $22.1 \pm 2.65$ |
| 1-NN | $704 \pm 0$ | $18.6 \pm 0$ |
| RNN | $563 \pm 39.5$ | $19.7 \pm 0.611$ |
| CALM-DT | $453 \pm 54.3$ | $15.3 \pm 0.999$ |
| $K$-NN ($K = 3$) | $368 \pm 0$ | $15.7 \pm 0$ |

*Table 9.* DT simulation comparisons of tritrophic algae-flagellate-rotifer population dynamics. Sorted by MSE. Averaged over 20 runs, presented with 95% CIs.

| Model | MSE ($\downarrow$) | MAE ($\downarrow$) |
|---|---|---|
| RNN | $0.156 \pm 8.01 \times 10^{-3}$ | $0.354 \pm 7.44 \times 10^{-3}$ |
| SINDy | $0.0265 \pm 0$ | $0.0994 \pm 0$ |
| 1-NN | $0.0178 \pm 0$ | $0.0748 \pm 0$ |
| $K$-NN ($K = 2$) | $0.0164 \pm 0$ | $0.0947 \pm 0$ |
| HDTwin | $2.89 \times 10^{-3} \pm 1.48 \times 10^{-3}$ | $0.0316 \pm 8.58 \times 10^{-3}$ |
| DyNODE | $2.57 \times 10^{-3} \pm 1.05 \times 10^{-3}$ | $0.0341 \pm 6.98 \times 10^{-3}$ |
| Transformer | $1.66 \times 10^{-3} \pm 7.54 \times 10^{-4}$ | $0.0283 \pm 5.70 \times 10^{-3}$ |
| CALM-DT | $3.87 \times 10^{-4} \pm 4.65 \times 10^{-5}$ | $0.0101 \pm 5.55 \times 10^{-4}$ |
| Constant | $1.00 \times 10^{-4} \pm 0$ | $5.40 \times 10^{-3} \pm 0$ |

sample efficiency of LLMs, and their bias towards simple numerical outputs (Gruver et al., 2024), does not excessively overfit or underperform on datasets where simple baselines excel beyond other deep learning approaches. CALM-DT is adept at simulating simple systems, as well as the more complex scenarios from §6.1.

## G. Simulation of Time to Key Events

To evaluate simulation accuracy from another angle, beyond raw MSE/MAE for simulated states, we examine the accuracy of critical event timing. Specifically, we examine the error in simulated vs real death times for untreated NSCLC patients.

**Setup.** We utilise the pharmacokinetic-pharmacodynamic model for NSCLC from Geng et al. (2017) to generate training and test data, although in this case we generate trajectories of patients who receive *no treatment* for 60 days. In this setup, patients experience extensive tumour growth and, using the threshold of 13cm tumour diameter to denote death, as is done in Geng et al. (2017), patient death can frequently occur within this 60-day untreated time-frame. As before, each DT method is provided with a 30-day input history and tasked with simulating the patient's trajectory for the subsequent 30 days. In Table 10 we report the error (in days) for simulated death compared to actual death.

**Takeaway.** CALM-DT demonstrates competitive performance in simulating accurate times to death for untreated NSCLC patients. It ranks as the third best method in this metric. This suggests CALM-DT has utility in forecasting clinically relevant event timings, complementing its raw simulation accuracy showcased in §6.1.

## H. Sample Selection Ablations on NSCLC

We conduct similar sample selection ablations to those in §6.2, now using the NSCLC dataset. We assess the contributions of the two main components of our sample selection scheme: (1) fine-tuning the encoders and (2) appending natural language

*Table 10.* Comparison of DT simulated time to death errors (days) for untreated NSCLC cancer patients. Averaged over five runs, with 95% CIs.

| Model | Time to death MAE |
|---|---|
| RNN | $10.25 \pm 0$ |
| Transformer | $6.32 \pm 1.72$ |
| HDTwin | $5.89 \pm 0.193$ |
| CALM-DT | $5.19 \pm 0.496$ |
| SINDY | $4.96 \pm 0$ |
| DyNODE | $4.91 \pm 0.146$ |

summaries of the time-series inputs provided to the encoders.

**Setup.** The experiments are conducted on the NSCLC dataset as described in §6.1. We vary the sample selection strategy for CALM-DT by enabling or disabling encoder fine-tuning and the inclusion of LLM-generated time-series summaries. The 'No fine-tuning/summary' condition uses `ModernBERT` (Warner et al., 2024) pre-trained encoders for $\phi_t$ and $\phi_c$ without any task-specific fine-tuning or summaries. 'No fine-tuning' uses the same pre-trained encoders with appended summaries to inputs at inference time. 'No summary' uses fine-tuned encoders but without appended summaries during training or inference. 'CALM-DT' represents the full proposed method with both fine-tuned encoders and appended summaries. Table 11 reports the simulation performances.

*Table 11.* CALM-DT NSCLC performance with different $C_f$ selection methods. Averaged over 10 runs, with 95% CIs. Ordered by MSE.

| $C_f$ Selection method | MSE ($\downarrow$) | MAE ($\downarrow$) |
|---|---|---|
| No summary | $101 \pm 8.72$ | $4.43 \pm 0.0748$ |
| No fine-tuning/summary | $89.3 \pm 19.1$ | $4.30 \pm 0.159$ |
| No fine-tuning | $85.7 \pm 7.03$ | $4.26 \pm 0.095$ |
| CALM-DT | $79.4 \pm 8.57$ | $4.28 \pm 0.157$ |

**Takeaway.** The results on the NSCLC dataset are broadly consistent with those observed for CF in §6.2, with the complete CALM-DT method, which incorporates both encoder fine-tuning and natural language summaries, achieving the best performance. Interestingly, and in contrast to the CF setting, fine-tuning the encoders without appending natural language summaries ('No summary') results in the poorest performance on this dataset. This may be due to the increased input length for the NSCLC data (30 time-steps), compared to CF data (3 time-steps). With longer sequences, the raw numerical data is inherently more complex. During fine-tuning on long, purely numerical sequences, the encoder faces a higher risk of latching onto spurious correlations or superficial numerical patterns that are not generalisable for effective retrieval. The appended summaries provide critical high-level semantic guidance in this case, helping the NLP encoder to discern meaningful long-range trends that it cannot extract from numerical inputs alone. Without this semantic anchor, the fine-tuning process for longer sequences might lead the encoder to learn less robust representations, hence the poor performance. This highlights a strong synergistic benefit between LLM-generated summaries and encoder fine-tuning for longer, more complex time-series inputs, where summaries become crucial for guiding the fine-tuning process towards learning genuinely informative features for retrieval rather than overfitting to spurious correlations in raw numerical data.

