# OpenReview forum: "Continuously Updating Digital Twins using Large Language Models"
_ICML.cc/2025/Conference — ICML 2025 poster_

### Official Review · Reviewer_dKC6 · 2025-03-07

**Overall Recommendation:** 3

**Summary:**

This paper proposes CALM-DT (Context-Adaptive Language Model-based Digital Twin), a novel digital twin framework that leverages large language models (LLMs) for simulation of dynamical systems.

**Claims And Evidence:**

Key claim: LLMs can serve as digital twins that continuously update without re-design or retraining.
Yes it is supported by the experiments

**Essential References Not Discussed:**

Not that I am aware of.

**Experimental Designs Or Analyses:**

Yes, the designs look valid.

**Methods And Evaluation Criteria:**

Yes the evaluations make sense

**Other Comments Or Suggestions:**

I don't have additional comments

**Other Strengths And Weaknesses:**

None

**Questions For Authors:**

Does the evaluation exceed the model's context window (i.e. 128k)? It seems not.

**Relation To Broader Scientific Literature:**

Introduces LLM-based, context-adaptive simulation, which relates to ongoing work in time-series forecasting via large language models but extends it by including actions/policies and retrieval-based updates.

**Theoretical Claims:**

The paper does not present new formal proofs (e.g., for convergence or bounds).

---

> ### Author Rebuttal · Authors · 2025-03-31
>
> Thank you for your thoughtful comments and suggestions. We give answers to the following:
>
> - (A) Is the context window exceeded?
> ---
>
> **(A) Is the context window exceeded?**
>
> Thank you for raising this point. You are correct in saying that the initial cystic fibrosis (CF) dataset we investigated does not exceed the LLM context limit (its combined textual representation of all samples is approximately 100k tokens) and it can therefore fit entirely within the LLM's context window. However, we have now **tested this scenario of including the entire training dataset in context as a further ablation**, and we find that it **leads to suboptimal results, significantly underperforming compared to our sample selection method** with context set size $c=5$ (See below table, which is our original Table 3 with the new ablation 'Whole dataset in context' added).
>
> Table 1: Sample selection ablations
> | Sample selection                 | MSE ($\downarrow$) | MAE ($\downarrow$) |
> |----------------------------------|--------------------|--------------------|
> | $\text{CALM-DT}_\text{Zero}$     | 74.816 $\pm$ 0.411 | 5.241 $\pm$ 0.021  |
> | $\text{CALM-DT}_\text{Random}$   | 73.761 $\pm$ 3.221 | 5.224 $\pm$ 0.100  |
> | $\text{CALM-DT}_\text{No train}$ | 67.055 $\pm$ 0.894 | 4.953 $\pm$ 0.040  |
> | Whole dataset in context         | 65.3 $\pm$ 3.90    | 4.81$\pm$0.102     |
> | $\text{CALM-DT}$                 | 55.336 $\pm$ 0.811 | 4.634 $\pm$ 0.045  |
>
> This finding **aligns with our existing context size ablation study in Section 6.2**, which demonstrates that optimal performance is achieved with $c=5$, and addition of more samples leads to performance degradation. These results further suggest that including excessive, potentially irrelevant samples degrades simulation performance. This observation is **supported by existing literature on LLM performance degradation with excessive context lengths** [1, 2].
>
> Additionally, we have now **conducted experiments on a larger dataset** to evaluate simulation performance for patients with Non-Small Cell Lung Cancer (NSCLC). We consider a dataset of 500 patients, each with 60 daily values recorded for the state variables 'Tumour volume' and 'Chemotherapy concentration,' as well as the action variables 'Chemotherapy dosage' and 'Radiotherapy dosage'. The size of each instance for this dataset is therefore substantially larger than in our previous CF experiment, making it **infeasible to tokenize the entire dataset and fit it within the LLM's context window** (the full dataset amounts to approximately 350K tokens). As a result, context sample selection is crucial in this setting.
>
> Our method demonstrates **strong performance on this dataset as well**, providing further evidence for the efficacy of our approach (See table below comparing 30 day simulation accuracy of CALM-DT with baseline models). CALM-DT achieves the highest accuracy in terms of MSE, and the second highest in terms of MAE, again using context set size of $c=5$, and GPT-4o as the base LLM.
>
> Table 2: 30 day simulation results on NSCLC dataset.
> | Model       | MSE ($\downarrow$)       | MAE ($\downarrow$) |
> |-------------|--------------------------|--------------------|
> | SINDy| $1.90 \times 10^4 \pm 0$ | $37.6 \pm 0$       |
> | Transformer | $116 \pm 52.7$           | $5.86 \pm 2.44$    |
> | RNN         | $115 \pm 4.35$           | $6.04 \pm 0.645$   |
> | DyNODE      | $104 \pm 63.6$           | $4.67 \pm 2.06$    |
> | HDTwin      | $80.6 \pm 11.8$          | $3.42 \pm 0.717$   |
> | CALM-DT     | $79.4\pm8.57$            | $4.28\pm0.157$     |
>
> Update: We have **run an additional sample selection ablation** for the CF experiment and included results in Section 6.2, evaluating the use of the **entire training dataset as a context set**. Furthermore, we have conducted an **additional experiment on a larger NSCLC dataset, which exceeds the LLM's context limit**. The results of this experiment are now included in Section 6.1.
>
> ---
>
> Thank you once again. We hope that we have addressed all your comments, and we greatly appreciate your feedback.
>
> ---
>
> [1] Liu, N. F., Lin, K., Hewitt, J., Paranjape, A., Bevilacqua, M., Petroni, F., and Liang, P. Lost in the middle: How language models use long contexts. Transactions of the Association for Computational Linguistics, 12:157–173, 2024
>
> [2] Li, T., Zhang, G., Do, Q.D., Yue, X. and Chen, W., Long-context llms struggle with long in-context learning, 2024. URL https://arxiv.org/abs/2404.02060.

---

### Official Review · Reviewer_SWpa · 2025-03-16

**Overall Recommendation:** 3

**Summary:**

This paper presents CALM-DT, a framework using large language models to create digital twins that can update continuously without redesign or retraining. Unlike traditional approaches, CALM-DT handles new variables and incorporates new information through in-context learning. Testing on cystic fibrosis patient data shows it outperforms existing methods and adapts seamlessly to changes like new treatments, making it valuable for dynamic real-world applications.

**Claims And Evidence:**

The authors only test their methodology in a single setting. For the general claims they are trying to make (proposing this as a general method), it seems important to have at least 3 or so settings. It seems hard to tell how generalizable some of the choices the authors make are across settings, and I think the effectiveness of certain choices would plausibly vary a lot by application setting.

Other than that, as far as I could tell, their claims and experiments were sound (with the exception of the things mentioned below).

**Essential References Not Discussed:**

I don't know references off the top of my head as this is not my main area of expertise, but mentioned something above that I could find with a quick search. My guess is that there are many other papers in that vein.

**Experimental Designs Or Analyses:**

Looking at the prompts, it's often not clear what is part of the actual prompt and what would be substituted with different values: e.g. in line 667, would there be an actual weight number provided? Would the `X` values be replaced with actual values later in the prompt? This seems inconsistent with providing the years. I think giving an example of the prompt in which all the variables are given explicitly (but having some way to highlight them as variables) would be helpful to better understand the setup.

**Methods And Evaluation Criteria:**

As mentioned above, I think the authors should have tested the method in more than one setting to make claims as general as they do (e.g. the title): as of right now, the work functions more so as a case study.

> However, since time-series data may differ significantly from typical NLP training distributions, we enhance each trajectory’s textual representation by appending an LLM-generated summary of its trends.

As far as I could tell, you do not validate with an ablation that appending an LLM-generated summary helps and how much. While I think this is probably quite context-specific (as I would guess many aspects of the method are). This seems especially important given that this is one of the points that the authors discuss as being an area of novelty ("Additionally, novelty arises in our sample-selection method, as we are the first to propose retrieval of time-series data by leveraging LLM generated summaries to enhance NLP encoder capabilities.").

**Other Comments Or Suggestions:**

Line ~380: "acros" -> "across"
Line 075: "accomodate" -> "accommodate"
Line 205: "flexbily" -> "flexibly"
Line ~375: "chane" -> "change"

Typo in one of the LLM prompts: "measurments" should be "measurements"

(About half of these typos were found by Claude.)

**Other Strengths And Weaknesses:**

Nothing I didn't already mention above!

**Questions For Authors:**

I think it would be interesting to ask the LLMs used in your experiments to hypothesize what "drug X" is referring to. I wouldn't be surprised if – when asked – they would be able to infer that it is Ivacaftor. I don't think this is a major methodological issue, but just wanted to point out that simple renaming may not be *entirely* sufficient to isolate the effect of your method, and this is something one could try to measure explicitly.

**Relation To Broader Scientific Literature:**

There is no mention in the paper of the relationship between digital twins and "world models". It's not clear to me what the difference is, and this makes it harder to assess the novelty of the paper is: that LLMs are relatively good world models is something that has already been studied quite a bit (even using in-context learning to simulate consequences of actions). For instance: [this paper](https://aclanthology.org/2025.coling-main.503.pdf), but there are likely many other works in this area. For example, based on the work linked, claims like the following seem likely incorrect to me:

> To the best of our knowledge, CALM-DT is the first proposed context-adaptive simulation method, that dynamically adjusts its knowledge and data base mid-generation.

As of right now my understanding is that the main novel contribution of this paper may be combining retrieval with using LLMs as world models (a relatively simple scaffolding change whose effectiveness is probably quite context-dependent), plus doing an in-depth case study in one specific application area.

Another area of claimed novelty may be the adaptation on the fly to new dynamics (e.g. new treatment options). However, this also likely has precedents in the literature: although most literature on world models is about static dynamics, I would be surprised if there wasn't any work that shows that e.g. LLMs are able to adjust their predictions on the fly when changing the rules of a game. A related line of work is that of LLMs as simulators of human behavior: even there, the premise is that LLMs would be able to predict human behavior in unseen situations, generalizing to new dynamics (e.g. [Generative Agent Simulations of 1,000 People](https://arxiv.org/abs/2411.10109)).

I think a much deeper engagement with analogues of this idea across the LLM literature is missing, as it seems essential to situate the contribution of the work. This is the main factor that makes me lean towards not recommending acceptance, paired with the limited scope of the evaluation.

**Theoretical Claims:**

I don't believe there are any theoretical claims in the paper.

---

> ### Author Rebuttal · Authors · 2025-03-31
>
> Thank you for your thoughtful comments and suggestions. We address the following:
>
> - (A) Additional settings
> - (B) Time-series summary ablation
> - (C) Prompts
> - (D) World models
> - (E) LLM thoughts on 'drug X'
>
> ---
> **(A) Additional settings**
>
> We evaluate CALM-DT on three additional datasets:
> 1) Non-Small Cell Lung Cancer (NSCLC) patients undergoing chemo- and radiotherapy (See Table 2 in our response to reviewer dKC6)
> 2) Di-trophic ecological system of hare and lynx population dynamics (See Table 1 in our response to reviewer Y6u8)
> 3) Tri-trophic ecological system of algae, flagellates, and rotifers (See Table 2 in our response to reviewer Y6u8)
>
> CALM-DT consistently outperforms, achieving the lowest MSE across all datasets and the lowest MAE on all but NSCLC. This shows the generalisability of CALM-DT and its capability to accurately model various dynamical systems.
>
> **_Update:_** We have **extended Section 6.1 with these 3 datasets**.
>
> ---
> **(B) Time series summary ablation**
>
> We conduct ablations for the LLM generated summaries for sample selection. We compare standard CALM-DT, using an encoder finetuned and tested with summaries appended to the time-series, to the use of finetuned encoders that are trained and tested without summaries on CF and NSCLC data. We also compare pretrained encoders, with/without summaries appended at inference time.
>
> [Link to table 1: CF results](https://anonymous.4open.science/r/CALM-DT-Rebuttals-65B2/CF_summary_ablation_results.png)
>
> [Link to table 2: NSCLC results](https://anonymous.4open.science/r/CALM-DT-Rebuttals-65B2/NSCLC_summary_ablation_results.png)
>
> On both datasets, appending LLM generated summaries allows the encoders to select better samples. This is true in the finetuned case, and the pretrained case.
>
> **_Update:_** We have **added an additional ablation to Section 6**, showing the importance of LLM generated summaries in sample selection.
>
> ---
> **(C) Prompts**
>
> Thank you for raising this point. In line 667, there would _not_ be an actual weight provided, as this is the _variable descriptions_ part of the prompt (in lines 664-671, the state/action variables are described). You are correct that the _X_ values in the supplied prompt would be replaced with actual values. We will make this clear by providing a full prompt in the appendix.
>
> **_Update:_** We have **added a full prompt to the appendix**.
>
> ---
> **(D) World models**
>
> We clarify that we see DTs as a class of world model with particular connotations. It is crucial that DTs (not necessarily world models in general) exhibit **continuous time dynamics** [1], to allow simulation of arbitrary length, and since a DT is tied to a specific individual system, it should **update its dynamics alongside it, as the system and its environment change** [2]. While world models can focus on more 'static' environments, **DTs must be designed to update**. Our method fits these DT requirements, given the adaptability which we show, and that LLMs can be prompted to simulate with arbitrary horizons and step sizes. We agree that a better contextualisation of this positioning is required, and, given space limitations, we will include an overview of relevant world models, especially those that incorporate LLMs, in our related works.
>
> We maintain that we are the first to propose context-adaptive simulations, enabled by sample retrieval, for LLM-based world models, which is essential to allow data-driven insights in unseen state-action pairs, without fine-tuning. While prior studies have indeed explored the use of LLMs as world models, they typically rely on static context/prompts during simulation, or require fine tuning to be performant. It is a further novelty that we show how LLM time-series in-context learning ability scales (Section 6.2).
>
> **_Update:_** We have **extended our related works** to build on the above comparison with world models.
>
> ---
> **(E) LLM thoughts on 'drug X'**
>
> Thank you for this insightful suggestion. We queried the LLM on drug X, and indeed Ivacaftor was one of its suggested options. To avoid this potential 'data leakage' we instead create a 'fake' drug name which is less clearly a placeholder than drug X. We replace 'drug X' with a more realistic, yet fake, CF drug name - 'Pulmurex' - and we see that it makes no appreciable difference to the post Ivacaftor results. Querying the LLM on Pulmurex, it no longer suggests it could be a placeholder for Ivacaftor.
>
> **_Update:_** We have **changed 'drug X' to a more plausible drug name** in our Ivacaftor experiments.
>
> ---
> Thank you once again. We hope that we have addressed all your comments, and we greatly appreciate your feedback.
>
> ---
> [1] Chen, H., Yang, J., Chen, J., Wang, S., Wang, S., Wang, D., Tian, X., Yu, Y., Chen, X., Lin, Y. and He, Y., 2024. Continuous-Time Digital Twin with Analogue Memristive Neural Ordinary Differential Equation Solver.
>
> [2] National Academies of Sciences. Foundational research gaps and future directions for digital twins. 2023

---

> > ### Comment · Reviewer_SWpa · 2025-04-03
> >
> > Thank you for responding to my points. I appreciate the additional experiments, and the discussion of world models. While I remain somewhat skeptical that you're the first to propose context-adaptive simulations for LLM-based world models, I agree that you're likely the first to do so with sample retrieval. I've raised my score.

---

> > > ### Author Response · Authors · 2025-04-09
> > >
> > > Thank you Reviewer SWpa, we are very grateful for your numerous helpful comments and continued engagement! Sample retrieval is a critical component of our context-adaptive simulation approach, as it enables CALM-DT to incorporate data-driven insights into simulation in an efficient and automatic way, by dynamically adjusting its context with the most relevant samples mid-simulation. Attempting to incorporate data-driven insights into simulation without sample retrieval, such as using a fixed context with a large number of samples included, leads to sub-optimal results (please see our response to Reviewer dKC6, part (A), where we show that including the entire CF dataset in context is less performant than our context-adaptive approach).
> > >
> > > We are glad that we have addressed your concerns, and we believe our manuscript has significantly improved as a result of these changes.

---

### Official Review · Reviewer_Y6u8 · 2025-03-25

**Overall Recommendation:** 4

**Summary:**

The paper addresses the challenge of maintaining the relevance of digital twins in dynamic environments where state/action variables and relevant information constantly change. The authors frame digital twinning as an in-context learning problem using LLMs. They propose CALM-DT, which uses fine-tuned encoders for sample retrieval, enabling accurate simulation across diverse state-action spaces during in-context learning.
The paper Identifies the limitations of existing DTs in dynamic environments (need for re-design or re-training) and its main contributions are: (1) Establishing design requirements for DTs for dynamic environments; (2) Demonstrating that LLMs can meet these requirements and proposing CALM-DT, which adapts to changes in its modeling environment without re-design or re-training; (3) Developing a simulation method that adjusts the information supplied to the LLM mid-generation to handle excessive context window lengths.
The paper empirically showing that CALM-DT outperforms existing DTs and can adapt to changes in modeling environments.

**Claims And Evidence:**

The paper’s claims are generally well-supported, with a couple points for improvement.

1. LLM Reliance- The performance of CALM-DT is heavily reliant on the underlying LLM. It would be beneficial to see a more thorough discussion of the limitations and potential risks associated with LLM biases, hallucinations, and failures.

2. More experiments- The experiments are primarily focused on modeling CF progression.  While this is a relevant and complex application, it is important to examine the generalization of CALM-DT to other domains. The authors should provide more discussion and evidence on how CALM-DT could be applied to different types of dynamic systems.

3. More evaluation metrics- The paper primarily uses MSE and MAE for evaluation. It would be helpful to include other evaluation measures relevant to the specific application such as predicting critical events.

**Essential References Not Discussed:**

None

**Experimental Designs Or Analyses:**

The paper is primarily centered on the CF domain. To strengthen the findings, the authors should broaden the empirical analysis to include additional domains and demonstrate CALM-DT's performance in more diverse settings. This would address concerns about the model's ability to generalize.
In the ablation study for sample selection, the paper compares different encoder-based methods. A relevant baseline would be to evaluate a selection strategy that uses another LLM (maybe a smaller one) to choose the context set $C_f$ based on the history, instead of relying on a trained encoder. This would provide insight into the effectiveness of the encoder approach compared to a more direct LLM-driven selection.

**Methods And Evaluation Criteria:**

The proposed method and evaluations are appropriate for addressing the problem. As noted- another evaluation metrics would be good.

**Other Comments Or Suggestions:**

None.

**Other Strengths And Weaknesses:**

The paper lacks clarity regarding how CALM-DT samples actions during the generation process. Key details about the action policy implementation are unclear and require further explanation.

**Questions For Authors:**

None.

**Relation To Broader Scientific Literature:**

This work is inspired by the broader context of LLMs and their ability to handle in-context learning and time-series data. These capabilities suggest that LLMs can function as adaptive data mechanisms, with applications in areas like medicine and finance.

**Theoretical Claims:**

No obvious problem.

---

> ### Author Rebuttal · Authors · 2025-03-31
>
> Thank you for your thoughtful comments and suggestions. We address the following:
>
> - (A) LLM reliance
> - (B) Further experiments
> - (C) Application-specific metrics
> - (D) LLM sample selection
> - (E) Actions
> ---
> **(A) LLM reliance**
>
> We agree it is important to elaborate on hallucinations, bias, and failures that can arise due to tokenization and formatting.
> - **Hallucination** can result in implausible patterns, spurious spikes/dips, or inconsistencies with known domain constraints. These can be difficult to predict, so careful analysis of outputs is critical.
> - **Biases** in training corpora can influence model predictions, potentially leading to disparities in simulations across populations.
> - Typical **tokenization** (e.g. BPE) can limit precision, as continuous variables are discretised, obliging caution in scenarios where high precision is necessary.
> - **Correctly structured outputs** cannot be guaranteed, although, in practice, we rarely experienced such issues. At times, textual explanations were included in outputs.
>
> **_Update:_** We have **extended Section 7** to expand on the above.
>
> ---
> **(B) Further experiments**
>
> We agree that demonstrating generalisation is crucial. Therefore, we evaluate CALM-DT on three additional datasets (as in [1]):
> 1) Non-Small Cell Lung Cancer (NSCLC) patients undergoing chemo- and radiotherapy (See Table 2 in response to reviewer dKC6)
> 2) Di-trophic ecological dynamics (Hare-Lynx population) (See Table 1 below)
> 3) Tri-trophic ecological dynamics (Algae-Flagellates-Rotifers population) (See Table 2 below)
>
> The results consistently demonstrate CALM-DT's superiority across these diverse settings.
>
> Table 1: Hare-Lynx 5 day simulation
> |Model|MSE($\downarrow$)|MAE($\downarrow$)|
> |---|---|---|
> |HDTwin|$1.11\times10^{4}\pm1.93\times10^4$|$29.6\pm9.21$|
> |Transformer|$2.52\times10^3\pm796$|$31.7\pm5.44$|
> |SINDy|$1.05\times10^3\pm0.00$|$26.5\pm0.00$|
> |DyNODE|$895\pm212$|$22.1\pm2.65$|
> |RNN|$563\pm39.5$|$19.7\pm0.611$|
> |CALM-DT|$453\pm54.3$|$15.3\pm0.999$|
>
> Table 2: Algae-flagellates-rotifers 5 day simulation
> |Model|MSE($\downarrow$)|MAE($\downarrow$)|
> |---|---|---|
> |RNN|$0.156\pm8.01\times10^{-3}$|$0.354\pm7.44\times10^{-3}$|
> |SINDy|$0.0265\pm0$|$0.0994\pm0$|
> |HDTwin|$2.89\times10^{-3}\pm1.48\times10^{-3}$|$0.0316\pm8.58\times10^{-3}$|
> |DyNODE|$2.57\times10^{-3}\pm1.05\times10^{-3}$|$0.0341\pm6.98\times10^{-3}$|
> |Transformer|$1.66\times10^{-3}\pm7.54\times10^{-4}$|$0.0283\pm5.70\times10^{-3}$|
> |CALM-DT|$3.87\times10^{-4}\pm4.65\times10^{-5}$|$0.0101\pm5.55\times10^{-4}$|
>
> **_Update:_** We have **extended Section 6.1** to include these 3 datasets.
>
> ---
> **(C) Application-specific metrics**
>
> We agree on the importance of application-specific metrics, and include an evaluation of critical event prediction. On NSCLC data (without treatments) we simulate time to patient death (set at tumour diameter of 13cm [2]). The table below shows the error (in days) for simulated death compared to actual death.
>
> |Model|Error|
> |---|---|
> |RNN|$10.25\pm0$|
> |Transformer|$6.32\pm1.72$|
> |HDTwin|$5.89\pm0.193$|
> |CALM-DT|$5.19\pm0.496$|
> |SINDY|$4.96\pm0$|
> |DyNODE|$4.91\pm0.146$|
>
> CALM-DT achieves third best. This, combined with the strong performance above, indicates the accuracy of CALM-DT's simulations overall and in capturing critical event timings.
>
> **_Update:_** We have **added an experiment on time-to-death simulation** to the appendix.
>
> ---
> **(D) LLM sample selection**
>
> Thank you for suggesting this insightful comparison with LLM-based sample selection. We test this on the CF dataset, querying an LLM, with the entire training dataset as context, for the top 5 samples for the current history. We see that CALM-DT is significantly better than this in both MSE and MAE.
>
> |Sample selection|MSE($\downarrow$)|MAE($\downarrow$)|
> |---|---|---|
> |LLM-based selection|65.6$\pm$2.19|$4.88\pm0.116$|
> |CALM-DT|55.3$\pm$0.811|4.63$\pm$0.045|
>
> Also, this is infeasible when training data exceeds the LLM context limit, e.g. for the NSCLC dataset. Our method for encoder finetuning with contrastive learning, determining positive/negative samples based on the simulation accuracy they induce, gives a useful signal to the encoder, resulting in better selections than an LLM.
>
> **_Update:_** We have **added this ablation** to Section 6.2.
>
> ---
> **(E) Actions**
>
> Consistent with [1], we consider a deterministic policy in testing. That is, at each simulation time point, we apply the same action that was applied to the test sample.
>
> **_Update:_** We **clarify action sampling in Section 6**.
>
> ---
> Thank you once again. We hope that we have addressed all your comments, and we greatly appreciate your feedback.
>
> ---
> [1] Holt, S., Liu, T. and van der Schaar, M., 2024. Automatically Learning Hybrid Digital Twins of Dynamical Systems
>
> [2] Geng, C., Paganetti, H. and Grassberger, C., 2017. Prediction of treatment response for combined chemo-and radiation therapy for non-small cell lung cancer patients using a bio-mathematical model

---

> > ### Comment · Reviewer_Y6u8 · 2025-04-04
> >
> > The authors have addressed all of my concerns including conducting additional experiments and ablations. Thank you for the effort, I'll raise my score.

---

> > > ### Author Response · Authors · 2025-04-09
> > >
> > > Thank you Reviewer Y6u8, we are very grateful for your numerous helpful comments and continued engagement! We are glad that we have addressed your concerns, and we believe our manuscript has significantly improved as a result of these changes.

---

### Official Review · Reviewer_Qbx4 · 2025-03-28

**Overall Recommendation:** 3

**Summary:**

This paper proposes a way to use a frozen LLM (e.g. GPT-4o) to construct an auto-regressive model of the temporal evolution of a few variables, like a medical patient's height, weight, and lung function measurement, in response to certain interventions (administration of medications). The main insight is that using an LLM instead of a hand-designed quantitative model allows the simulator to adapt to changes in the state or action space without requiring new iterations of manual modeling.  They devise a scheme for retrieving similar trajectories from a dataset for use in in-context learning, based on fine-tuning encoders using a contrastive loss that is itself based on the LLM. They show that their method forecasts lung function measurements with less error than some other machine learning based approaches on a cystic fibrosis dataset (CF), and they show an example of how their methodology adapts to introduction to a new CF treatment without requiring retraining.

**Claims And Evidence:**

Their methodology does seem more capable in principle of adapting to changes in the state and action space without expert human modeling intervention, and this feature makes this research direction promising. But I do not find their demonstration of the method's in-context adaptation to a new Ivacaftor action (Section 6.3) very compelling. In particular, it is concerning that the mean absolute error increased when trajectories that include Ivacaftor were introduced (Table 4, MAE, K update vs K + D update). The amount of variability in Table 4 and Table 5 also makes it difficult to evaluate the (i) the capability of the methodology to adapt to a new action (Table 4) and (ii) the capability of the model to learn from additional data (Table 5).

Also, they claim (Table 1) that their method "allows uncertainty in simulation" (presumably implemented by running simulations multiple times), but there is no analysis of the variability in model outputs and so there's no reason to believe that this variability is meaningful (especially since the simulation kernel LLM was not trained using a prediction loss that would in theory relate to the distribution in outputs to the epistemic uncertainty in the predictions).

**Essential References Not Discussed:**

See "Relation to Broader Scientific Literature" above.

**Experimental Designs Or Analyses:**

No.

**Methods And Evaluation Criteria:**

The dataset they use does seem suitable for demonstrating their methodology. However, given the lack of a mechanistic principle for this methodology (i.e. the model does not seem to be trained to minimize prediction error), more evaluation is needed to understand whether the predictions are meaningful. I would have liked to (i) see simpler baselines in Table 2 (constant prediction, K-Nearest-Neighbors), and (ii) randomly selected plots of concrete predictions of this model (with multiple simulations to convey something about the variability of the outputs and any LLM-related artifacts) compared to the ground truth values and baseline models.

In order to better convey whether this evaluation is meaningful, the paper could compare the prediction error against state-of-the-art longitudinal models in the evaluation domain (not only against generic machine learning models or models in the digital twins literature). For example, I would like to know how the prediction error compares to that of this recent paper: "Predicting lung function decline in cystic fibrosis: the impact of initiating ivacaftor therapy", Respiratory Research 2024. I understand that a systematic comparison to this model might not be possible due to data limitations, but do the predictions have comparable levels of error? How might the types of errors produced by this method differ from those produced by a hand-designed model? Yes, the paper's proposed approach doesn't require human modeling, but the paper should investigate the tradeoffs of the proposed methodology with alternative approaches.

**Other Comments Or Suggestions:**

No.

**Other Strengths And Weaknesses:**

I think that the capability of LLMs to adapt to changes in state and action space parameterizations is an important idea, and I think this paper identified a suitable setting to investigate these ideas. The paper is also clearly written. I have concerns about the significance of the paper, due to a lack of connection to state-of-the-art models in the evaluation domain (which is biostatistics / medical informatics).

**Questions For Authors:**

1. How does your model's prediction accuracy compare to that of expert models like the Respiratory Research paper I cited above? If a comparison is not possible, are there any other works outside of the digital twins literature that can help ground these results and help readers determine the potential impact of this approach to this domain?

2. Is the model's uncertainty (probability distribution on forecasts) meaningful or interpretable in any way? For example, are there multiple modes of disease progression that it captures?

3. What is the tokenization that the LLM emits for quantitative predictions? Does each digit typically get one token? Do the forecasts have any artifacts related to tokenization or use of language modality for prediction?

**Relation To Broader Scientific Literature:**

As mentioned above, the paper does not adequately evaluate the proposed method relative to the broader scientific literature. For example, it is not clear how this methodology compares to state-of-the-art models from the biostatistics literature, like "Predicting lung function decline in cystic fibrosis: the impact of initiating ivacaftor therapy", Respiratory Research 2024. Clearly, the proposed methodology requires less manual effort, but at what cost?

**Theoretical Claims:**

No.

---

> ### Author Rebuttal · Authors · 2025-03-31
>
> Thank you for your thoughtful comments and suggestions. We address the following:
>
> - (A) Ivacaftor experiments
> - (B) Uncertainty
> - (C) Simple baselines
> - (D) Domain-specific model
> - (E) Tokenization
>
> ---
> **(A) Ivacaftor experiments**
>
> We agree that more robust evidence is necessary for demonstrating our method's adaptability. Initially, limited sample size and few iterations contributed to high variance in results. To rectify this, we've increased test set size to 50 patients and performed 20 simulation iterations, resulting in clearer outcomes.
>
> Table 1: Adapting to new action
> |Method|MSE($\downarrow$)|MAE($\downarrow$)|
> |---|---|---|
> |No update|59.7$\pm$0.472|4.95$\pm$0.0166|
> |$K$ update|54.4$\pm$ 0.472|4.75$\pm$0.0242|
> |$K$ + $D$ update| 50.4 $\pm$ 0.477| 4.28 $\pm$ 0.0251|
>
> Table 2: Learning from new data
> |Context future|MSE($\downarrow$)|MAE($\downarrow$)|
> |---|---|---|
> |One year|50.4$\pm$0.477|4.28$\pm$0.0251|
> |Two years|50.0$\pm$1.00|4.29$\pm$0.0468|
> |Three years|47.8$\pm$1.01|4.19$\pm$0.0523|
>
> We now have more robust and interpretable results. Table 1 demonstrates that updating K and incorporating Ivacaftor trajectories into D both significantly improve simulation accuracy, as evidenced by substantial reductions in MSE and MAE, illustrating CALM-DT's capability to adapt effectively to new actions. Similarly, in Table 2, we see that CALM-DT can learn from new data, as there is a clear improvement in simulation from using 1 to 3 years of context data.
>
> **_Update:_** We have **improved our experimental set-ups in Sections 6.3 and 6.4**, giving **more clear takeaways**.
>
> ---
> **(B) Uncertainty**
>
> We acknowledge the need for deeper analysis of simulation uncertainty. To address this, we provide illustrative plots demonstrating variability across simulations of cancer tumours (please see our response to reviewer dKC6 for context) under certain settings (Zero, one, and five shot) in our anonymous repository [here](https://anonymous.4open.science/r/CALM-DT-Rebuttals-65B2). The dark line is the true trajectory, and the grey lines are individual simulations. With more context, simulation variance decreases, thus reflecting meaningful epistemic uncertainty. Quantitatively, average simulation variance across the NSCLC dataset decreases from 2.79 (zero-shot) to 1.58 (five-shot)
>
> Also, we note that in our encoder finetuning, we use the Continuous Ranked Probability Score (CRPS) to guide the selection of positive and negative samples for contrastive learning. Since CRPS measures how well the simulated distribution aligns with the true outcome, this training procedure encourages selection of related samples that calibrate the uncertainty of the LLM well.
>
> **_Update:_** We **include illustrative plots of CALM-DT simulations**, showing how context reduces simulation uncertainty.
>
> ---
> **(C) Simple baselines**
>
> We appreciate your suggestion to benchmark our method against simpler baselines. We now include constant prediction and nearest-neighbour baselines for the CF data:
>
> |Model|MSE($\downarrow$)|MAE($\downarrow$)|
> |---|---|---|
> |Constant prediction|86.8|5.84|
> |1-NN| 107|6.96|
> |K-NN (K=12)|65.1|5.47|
>
> CALM-DT remains the significantly best performing simulation method in both MSE and MAE.
>
> **_Update:_** We have **expanded our baseline methods** in Section 6.1 to include constant prediction, 1-NN, and K-NN.
>
> ---
> **(D) Domain-specific model**
>
> We agree that it is very useful to give context of SOTA domain-specific methods. The provided paper reports a RMSE for _FEV1pp_ (which we also simulate) of 6.78 over 6 months for patients on Ivacaftor. From our Ivacaftor experiment, using our most performant setting, CALM-DT's _FEV1pp_ RMSE for 1 year is 8.11, and for 3 years is 8.58 (note we can only give yearly errors, as this is the granularity of the UK CF registry data). Using this domain-specific model as an upper bound on performance, we see that CALM-DT performs relatively well, especially given its low expertise requirements and seamless adaptability to changes in environment. We believe this added comparison strengthens the significance of our method.
>
> **_Update:_** We have **added a discussion in Section 6.1** comparing our results to the provided reference.
>
> ---
> **(E) Tokenization**
>
> We agree on the importance of considering tokenization impacts. Since we propose a simulation method that is compatible with any LLM, tokenization is not bespoke for CALM-DT. Although tokenization in GPT-4o (the model we use in experiments) is undisclosed, typical tokenization approaches (e.g., BPE) inherently limit numerical precision. Continuous variables are discretised into tokens, and so we suggest caution in scenarios where high precision is necessary.
>
> **_Update:_** We **extend our discussion in Section 7** to elaborate on LLM limitations.
>
> ---
>
> Thank you once again. We hope that we have addressed all your comments, and we greatly appreciate your feedback.

---

### Decision · Program_Chairs · 2025-05-01

**Decision:**

Accept (poster)

**Comment:**

This paper introduces an approach for leveraging LLMs to create digital twins (DT) capable of continuous updates without retraining. The core idea of framing DT adaptation as an in-context learning problem addresses a significant limitation in traditional DT methodologies and is conceptually strong. The authors demonstrate their method's ability to adapt to new actions and incorporate new data within the Cystic Fibrosis case study, showcasing the potential of this approach.
Reviewers raised some concerns regarding the initial empirical evaluation beyond the primary dataset and the comparison to domain-specific state-of-the-art models. The authors added experiments and discussion in rebuttal addressing these points. It is important to note that the method's performance is inherently tied to the capabilities and limitations of the underlying LLM, including potential biases and computational overhead. The authors provided substantial revisions and clarifications during the rebuttal phase, which satisfied several reviewers and led to score increases. Overall, the paper presents a promising direction for dynamic digital twins. Further validation across diverse domains could strengthen the claims.